# High-fidelity realization of the AKLT state on a NISQ-era quantum processor

Tianqi Chen,[1, *] Ruizhe Shen,[2, †] Ching Hua Lee,[2, 3, ‡] and Bo Yang[1, 3, §]

[1]*School of Physical and Mathematical Sciences, Nanyang Technological University, Singapore 639798*
[2]*Department of Physics, National University of Singapore, Singapore 117542*
[3]*Institute of High Performance Computing, A*STAR, Singapore 138632*

The AKLT state is the ground state of an isotropic quantum Heisenberg spin-1 model. It exhibits an excitation gap and an exponentially decaying correlation function, with fractionalized excitations at its boundaries. So far, the one-dimensional AKLT model has only been experimentally realized with trapped-ions as well as photonic systems. In this work, we successfully prepared the AKLT state on a noisy intermediate-scale quantum (NISQ) era quantum device for the first time. In particular, we developed a non-deterministic algorithm on the IBM quantum processor, where the non-unitary operator necessary for the AKLT state preparation is embedded in a unitary operator with an additional ancilla qubit for each pair of auxiliary spin-1/2's. Such a unitary operator is effectively represented by a parametrized circuit composed of single-qubit and nearest-neighbor $CX$ gates. Compared with the conventional operator decomposition method from Qiskit, our approach results in a much shallower circuit depth with only nearest-neighbor gates, while maintaining a fidelity in excess of 99.99% with the original operator. By simultaneously post-selecting each ancilla qubit such that it belongs to the subspace of spin-up $|\uparrow\rangle$, an AKLT state can be systematically obtained by evolving from an initial trivial product state of singlets plus ancilla qubits in spin-up on a quantum computer, and it is subsequently recorded by performing measurements on all the other physical qubits. We show how the accuracy of our implementation can be further improved on the IBM quantum processor with readout error mitigation.

## I. INTRODUCTION

The current age has witnessed tremendous progress in the quantum simulation of novel many-body phenomena [1–6]. In particular, there has been intense recent focus on using noisy intermediate-scale quantum (NISQ)-era [7] quantum computers to assist in large-scale tasks with the goal of eventual quantum supremacy [8, 9]. Among them, programmable digital quantum computers have so far been successfully used for the implementation and study of discrete time crystals (DTC) [10, 11], quantum chemistry problems with Hartree-Fock methods [12], fractional quantum Hall states [13, 14], spin chain dynamics [15, 16], interacting topological lattice models [17, 18], many-body localization [19], lattice gauge theory [20] and quantum spin liquid states [21]. These examples in general involve (but are not limited to) three categories of usage of quantum computers for condensed matter physics: time evolution, ground state search and state preparation. Such efforts are made with the goal of overcoming major drawbacks in current numerical approaches. These include the exponential "curse" of exact diagonalization (ED) [22], the sign problem in Fermionic quantum Monte Carlo simulations [23], and the rapid growth of entanglement in tensor network states [24, 25].

At the current juncture, there are still limitations and challenges in using NISQ-era quantum computers for large scale simulations. Some major issues include large circuit depth, low gate fidelity, and thermal noise from the execution of the quantum circuit [26]. In response, many classical algorithms and approaches based on matrix product state (MPS) have been recently proposed for state preparation [27–29]. Another challenge, which would be present even for a perfect quantum computer, is that state preparation is a fundamentally non-unitary process which requires the implementation of non-unitary operators. Progress has lately been made through imaginary time evolution approaches combined with variational algorithms [30, 31], or through constructing a deterministic measurement operator [32, 33]. However, these techniques may not always be practical for NISQ-era quantum processors such as the IBM Q system due to short qubit coherence times. In general, various requisite operators cannot be directly decomposed into the fundamental unitary gates on NISQ-era quantum computers, posing difficulties for existing schemes for state preparation.

In this work, we propose an algorithm and demonstrate the preparation of a particular type of quantum many-body state, the so-called Affleck, Kennedy, Lieb, and Tasaki (AKLT) state [34, 35], on NISQ-era quantum computers. As a type of Valence-Bond-Solid (VBS) state [35], it is the exact ground state of the spin-1 AKLT model, which is the paradigm of a strongly correlated symmetry protected topological (SPT) phase with a Haldane gap [36] and fractional excitations at its boundaries [34, 37, 38]. SPT phases of matter received much attention recently on quantum computers [17, 39–44], and the two-dimensional generalization of the AKLT model on a trivalent lattice is proposed to be a universal resource [45, 46] for measurement-based quantum compu-

* tianqi.chen@ntu.edu.sg
† ruizhe20@u.nus.edu
‡ phylch@nus.edu.sg
§ yang.bo@ntu.edu.sg

arXiv:2210.13840v2 [quant-ph] 9 Feb 2023

tation [47–49]. So far, the 1D AKLT state has been experimentally realized and characterized on photonic implementations [50] using cluster states [51] and in trapped ions [52]. Recently, we notice that there have been much efforts to construct the VBS state, including in particular the AKLT state in 1D with measurement assisted preparation [53], and in 2D with a post-selection algorithm [54]. With the usage of tensor network states, both 1D and 2D AKLT states can be prepared adiabatically [55]. For our work, instead of performing the variational searching of the AKLT state as the ground state of the spin-1 AKLT model [56], we show that the AKLT state can be obtained by evolving from a trivial initial product state composing of a chain of singlets. On a NISQ-era quantum computer (e.g., IBMQ), the main challenge is the non-unitarity of the state preparation, and our new approach is based on augmenting the non-unitary subspace with additional ancilla qubits, such that an effectively non-unitary operator can be realized through measurement-based post-selection. This allows us to implement non-unitary operators with unitary gates, achieving the simultaneous non-unitary projection on every site of an initial product state made up of a chain of singlets. For an efficient quantum circuit realization of this unitary operator, another matrix product state (MPS)-based algorithm on a classical computer is used to transform the operator into a parametrized circuit via variational optimization [17, 57–60]. Most recently, MPS-based algorithms have been applied for the investigations of translational-invariant systems [61, 62]. Compared with other recent AKLT state preparation methods [53–55], our approach only requires nearest-neighbor $CX$ gates, and the full circuit that prepares the AKLT state is much shallower than that from Qiskit's default isometry decomposition method [63]. Also, the evolution from the initial state has only one step, and it does not require any mid-circuit measurements on IBM Q [64].

This paper is organized as follows. First, in Sec. II, we introduce the AKLT model and its ground state, i.e. the AKLT state. Sec. III discusses the details of the approach used in this work to prepar the AKLT state, which includes transforming the projection operator into a unitary one, a variational parametrized circuit for the three-qubit operator, and post-selection of the results. Sec. IV presents the characterization of AKLT states for $L = 2, 3, 4$ and $5$ on IBMQ devices, and discusses various factors which could affect the fidelity of the prepared state. Finally, we highlight the conclusion of this work in Sec. V.

## II. THE AKLT STATE

Below, we briefly introduce the AKLT state. Consider a 1D spin chain with $2L$ spin-1/2s, grouped into pairs of adjacent spins as illustrated in FIG. 1(a). In general, each pair of adjacent spin-1/2s either forms a spin-0 singlet state $(|\uparrow\downarrow\rangle - |\downarrow\uparrow\rangle)/\sqrt{2}$, or one of the three symmetric

states

$$|+\rangle = |\uparrow\uparrow\rangle \tag{1}$$
$$|O\rangle = 1/\sqrt{2}\,(|\uparrow\downarrow\rangle + |\downarrow\uparrow\rangle)$$
$$|-\rangle = |\downarrow\downarrow\rangle$$

which spans the spin-1 subspace. To construct the AKLT state, we first project onto the spin-1 subspace of each pair of adjacent spin-1/2s [circled in FIG. 1(a)], such that we obtain an effective chain of $L$ spin-1s.

Before any constraints are applied, each pair of adjacent spin-1s can have a total spin of $S = 0, 1$ or $2$. The AKLT state is the unique state satisfying the constraint that every pair of adjacent spin-1s (i.e. the four consecutive spin-1/2s in two adjacent circles) is allowed to have a total spin of $S = 0$ or $1$, but not $2$. In terms of the constituent spin-1/2s, this is equivalent to the constraint that each spin-1/2 forms a (spin-0) singlet with another spin-1/2 from an adjacent spin-1 pair, as illustrated in FIG. 1(a). This would be the picture that our AKLT state algorithm is based on - we shall first prepare the spin singlets, and next project spin-1/2 pairs connected to adjacent singlets onto their total $S = 1$ subspace.

The above spin chain picture can be recast as an MPS representation of the AKLT state $|\psi\rangle$, for both periodic and open boundary conditions (PBCs and OBCs):

$$|\psi\rangle_{\text{PBC}} = \sum_{\boldsymbol{\sigma}} \text{Tr}\,[A^{\sigma_1} A^{\sigma_2} \cdots A^{\sigma_L}] \,|\sigma_1 \sigma_2 \cdots \sigma_L\rangle, \tag{2}$$

$$|\psi\rangle_{\text{OBC}} = \sum_{\boldsymbol{\sigma}} \left[ b_A^{l\,T} A^{\sigma_1} A^{\sigma_2} \cdots A^{\sigma_L} b_A^r \right] |\sigma_1 \sigma_2 \cdots \sigma_L\rangle, \tag{3}$$

where $\sigma_i \in \{+, O, -\}$ labels the $i$-th spin-1 basis state, with corresponding MPS matrices $A^\sigma$ given by

$$A^+ = +\sqrt{\frac{2}{3}}\tau^+, A^0 = -\sqrt{\frac{1}{3}}\tau^z, A^- = -\sqrt{\frac{2}{3}}\tau^-, \tag{4}$$

$\tau^z$ and $\tau^\pm = \tau^x \pm i\tau^y$ spanning the set of Pauli matrices [65, 66]. Since $(\tau^\pm)^2 = 0$, this matrix representation keeps track of the AKLT constraint that two adjacent spin-1s do not add up to total spin $S = 2$. Under PBCs, there has to be an equal number of $|+\rangle$ and $|-\rangle$ in $|\sigma_1\sigma_2\cdots\sigma_L\rangle$, as enforced by the trace operator Tr. Under OBCs, which is the more convenient scenario for implementation on the quantum processor [see Fig. 4], we will have to fix the end spins – in the above, we have chosen these boundary vectors to be $b_A^l = \begin{pmatrix} 1 & 0 \end{pmatrix}^T$, and $b_A^r = \begin{pmatrix} 0 & 1 \end{pmatrix}^T$, up to a normalization factor. This means that both boundary spins are fixed as spin up, which is the same as the MPS representation described in Ref. [67]. As is shown below in Fig. 2, this choice is of convenience for our implementation, as all qubits are initialized as spin up on the IBM Q system. For this choice, there is necessarily one more $|+\rangle$ compared to the

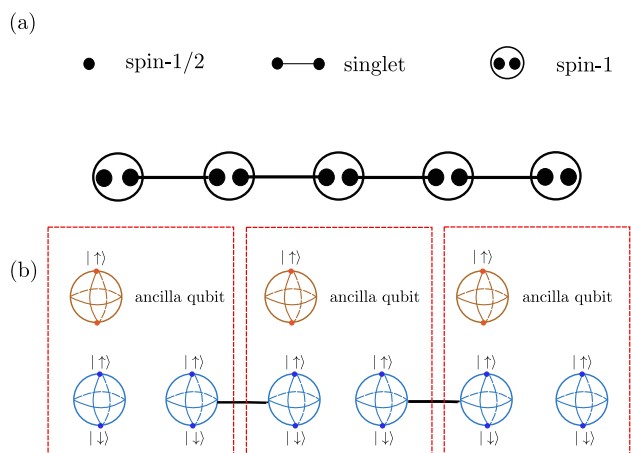

(a)

(b)

FIG. 1. **Structure of the AKLT state.** (a) The AKLT state with open boundary conditions (OBCs) with $L = 5$: solid lines connect two spin-1/2 qubits, forming singlet states. Each pair of spin-1/2s (circled) from two consecutive singlets are projected via $\hat{\mathcal{P}}$ onto their total spin $S = 1$ sector, via Eq. (6). (b) With the help of ancilla qubits (brown spheres), the non-unitary projectors $\hat{\mathcal{P}}$ can be embedded in unitary operators $\hat{\mathcal{U}}$ acting on three qubits (red dashed square): the two spin-1/2s (blue spheres) plus the ancilla qubit. The solid black line connecting two physical qubits forms a singlet. The actual physical embedding into IBM quantum processor qubits is shown Sect. III D.

number of $|-\rangle$. The explicit forms of $|\psi\rangle_{\mathrm{PBC}}$ and $|\psi\rangle_{\mathrm{OBC}}$ are given in Appendix C for $L = 2$ and $L = 3$.

Although we shall directly prepare the AKLT state through an MPS (Eqs. 2 and 3) quantum circuit, we note in passing that the AKLT state can also be obtained as the unique zero ground state [34, 35] of the following projection operator:

$$\hat{P}_{S=2} = \sum_{i=1}^{L-1} \left[ \mathbf{S}_i \cdot \mathbf{S}_{i+1} + \frac{1}{3} \left( \mathbf{S}_i \cdot \mathbf{S}_{i+1} \right)^2 + \frac{2}{3} \right], \quad (5)$$

which projects onto the total spin $S = 2$ sector in all pairs of adjacent spin-1s. It can be derived [34] by considering the total spin operator $(\mathbf{S}_i + \mathbf{S}_{i+1})^2$ with eigenvalues proportional to $S(S+1)$, where $\mathbf{S}_i$ and $\mathbf{S}_{i+1}$ are the spin-1 operators of adjacent spin-1s.

## III. PREPARING THE AKLT STATE ON A QUANTUM COMPUTER

### A. Implementing local projections within unitary operators

The preparation of the AKLT state on a quantum circuit crucially requires non-unitary operators for projecting onto the spin-1 subspace of each adjacent spin-1/2 pair [FIG. 1(a)]. Inspired by the techniques introduced in Refs. [29, 68, 69], we develop an approach for preparing the AKLT state by embedding the projection operator on each spin-1/2 pair into a 3-qubit unitary operator that admits an additional ancilla qubit. By subsequently projecting the ancilla qubit onto a chosen state $|\uparrow\rangle$ by post-selection [see FIG. 1(b)], we can realize the non-unitary $S = 1$ projection on the two spin-1/2s. This approach allows us to prepare the AKLT state according to the MPS formalism given in [70].

Explicitly, as shown in FIG. 1 (a), a local spin-1 in the bulk, which forms one "site" of an AKLT chain in OBCs, is built from a pair of adjacent spin-1/2 through the projection operator

$$\hat{\mathcal{P}} = |+\rangle \langle+| + |O\rangle \langle O| + |-\rangle \langle-| \qquad (6)$$
$$= \begin{pmatrix} 1 & 0 & 0 & 0 \\ 0 & 1/2 & 1/2 & 0 \\ 0 & 1/2 & 1/2 & 0 \\ 0 & 0 & 0 & 1 \end{pmatrix},$$

expressed in the spin-1/2 basis $\{|\uparrow\uparrow\rangle, |\uparrow\downarrow\rangle, |\downarrow\uparrow\rangle, |\downarrow\downarrow\rangle\}$, with $|+\rangle = |\uparrow\uparrow\rangle$, $|O\rangle = 1/\sqrt{2}(|\uparrow\downarrow\rangle + |\downarrow\uparrow\rangle)$ and $|-\rangle = |\downarrow\downarrow\rangle$ defined as before. Note that the spin-1/2s are the physical degrees of freedom on a quantum processor, even though they are often referred to as virtual spins in the AKLT literature.

This spin-1 projector of Eq. (6) is non-unitary as $\hat{\mathcal{P}}\hat{\mathcal{P}}^\dagger \neq I$, which is not possible to be directly implemented on a quantum computer such as IBM Q. To realize it in a quantum circuit, we embed it in a 3-qubit unitary operator $\hat{\mathcal{U}}$ which takes the form

$$\hat{\mathcal{U}} = \left[ \begin{array}{c|c} \hat{\mathcal{P}} & \hat{\mathcal{Q}} \\ \hline \hat{\mathcal{Q}} & \hat{\mathcal{P}} \end{array} \right] \qquad (7)$$

in the product basis of the ancilla qubit and the two spin-1/2 qubits. Here, we stipulate

$$\hat{\mathcal{Q}} = \begin{pmatrix} 0 & 0 & 0 & 0 \\ 0 & 1/2 & -1/2 & 0 \\ 0 & -1/2 & 1/2 & 0 \\ 0 & 0 & 0 & 0 \end{pmatrix}, \qquad (8)$$

such that $\hat{\mathcal{U}}$ is unitary, i.e. $\hat{\mathcal{U}}^\dagger \hat{\mathcal{U}} = I$. As both $\hat{\mathcal{P}}$ and $\hat{\mathcal{Q}}$ are symmetric, it is easy to verify that $\hat{\mathcal{P}}^2 + \hat{\mathcal{Q}}^2 = I_{4\times 4}$, and $\hat{\mathcal{Q}}\hat{\mathcal{P}} + \hat{\mathcal{P}}\hat{\mathcal{Q}} = 0_{4\times 4}$.

For this three-site subsystem consisting of two original spin chain qubits and an ancilla qubit, we examine input states of the form

$$|\psi\rangle = |\uparrow\rangle \otimes |\phi\rangle = \begin{pmatrix} 1 \\ 0 \end{pmatrix} \otimes |\phi\rangle, \qquad (9)$$

where $|\uparrow\rangle$ represents an ancilla qubit in the spin-up state, and $|\phi\rangle$ represents the two adjacent qubits which are paired as a singlet. Applying Eq. (7) to the above three-qubit state, we have

$$\hat{\mathcal{U}} |\psi\rangle = \begin{pmatrix} \hat{\mathcal{P}} |\phi\rangle \\ \hat{\mathcal{Q}} |\phi\rangle \end{pmatrix} = |\uparrow\rangle \otimes \hat{\mathcal{P}} |\phi\rangle + |\downarrow\rangle \otimes \hat{\mathcal{Q}} |\phi\rangle. \qquad (10)$$

Therefore, it is clear that after projecting the output ancilla qubit onto $|\uparrow\rangle$, say via post-selection, the target state $|\phi\rangle$ is indeed acted on by the nonunitary projector $\hat{\mathcal{P}}$ i.e.

$$\langle\uparrow|\,\hat{\mathcal{U}}\left(|\uparrow\rangle\otimes|\phi\rangle\right)=\hat{\mathcal{P}}|\phi\rangle. \tag{11}$$

The above technique contains only one single-step evolution that does not require any mid-circuit measurement for the preparation of the AKLT state [64], which provides a new approach towards embedding a non-unitary projection operator $\hat{\mathcal{P}}$ into a unitary operator $\hat{\mathcal{U}}$ for further decomposition into basis gates on a quantum computer, which will be discussed shortly. We also remark that our approach can be used to prepare the AKLT state under both OBCs to PBCs, although the PBC case requires a quantum device geometry such that a closed loop of qubits exists, and are accompanied by appropriately located branches functioning as ancilla qubits [see Fig. 4] [71].

## B.  Quantum circuit implementation

We break up the preparation of the MPS-based AKLT state into two steps, as sketched in FIG. 2. We first prepare the paired singlet states [two solid dots connected with a solid line in Fig. 1(a)] as initial states through the combination of $X$ gates, a Hadamard gates and a $CX$ gate (see notations in Qiskit [63]), which corresponds to the operations to the left of the red dashed line in FIG. 2. An $X$ gate is essentially a Pauli-$X$ operator, and a Hadamard gate $H$ maps $|\uparrow\rangle\,[|\downarrow\rangle]$ to $(|\uparrow\rangle+|\downarrow\rangle)/\sqrt{2}\,[(|\uparrow\rangle-|\downarrow\rangle)/\sqrt{2}]$:

$$X=\begin{pmatrix}0&1\\1&0\end{pmatrix}, H=\frac{1}{\sqrt{2}}\begin{pmatrix}1&1\\1&-1\end{pmatrix}, \tag{12}$$

while a $CX$ gate is a two-qubit controlled-X gate which performs a Pauli-$X$ operation on the target qubit whenever the control is in state $|\downarrow\rangle$.

$$CX=\begin{pmatrix}1&0&0&0\\0&1&0&0\\0&0&0&1\\0&0&1&0\end{pmatrix}. \tag{13}$$

which is expressed in the same basis as in Eq. (6). Next, we perform the $S=1$ projections $\hat{\mathcal{P}}$ on spins-1/2 pairs from adjacent singlets, which is undertaken by the unitary operation $\hat{\mathcal{U}}$ of Eq. (7). The ancilla qubits associated with the first, second, third etc pairs are labeled "$q_0$", "$q_3$" and "$q_6$" etc. in FIGs. 1(b) and 2.

FIG. 2 shows the circuit structure for a small illustrative system with $L=3$; we point out that the aforementioned procedure can be extended to arbitrarily large system sizes, as one can apply the unitary operator $\hat{\mathcal{U}}$ simultaneously to all corresponding sites, i.e.

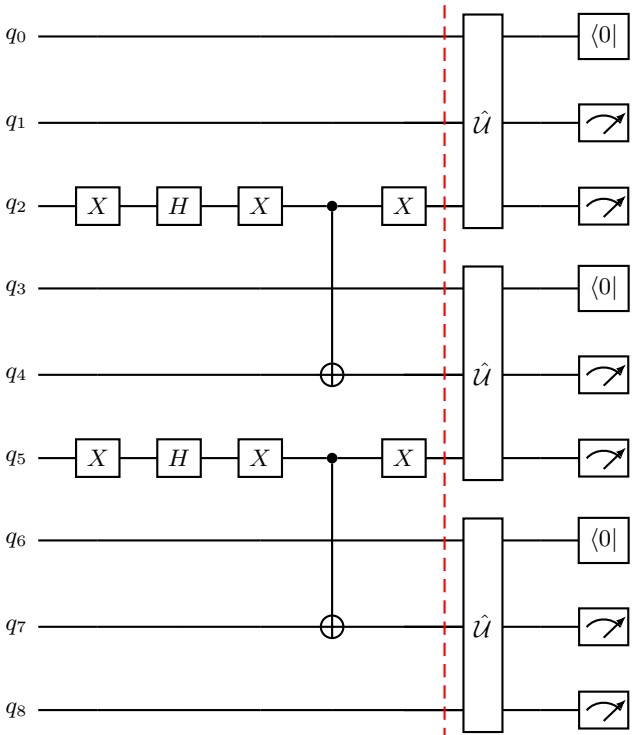

FIG. 2. **Illustrative quantum circuit for the preparation of a $2L=6$-qubit AKLT state with OBC.** The roles of the 9 qubits $q_0$ to $q_8$ are given in Fig. 1(b), with $L$ ancilla qubits $q_0, q_3, q_6$ and the remain $2L$ qubits representing the spin-1/2 chain. Qubits $q_1$ and $q_8$ are boundary qubits while qubits $q_2, q_4$ and $q_5, q_7$ pair up as singlets. The state preparation consists of two steps, as separated by the red dashed line: First, the initial state consisting of $L-1=2$ singlet pairs is initialized as shown to the left of the red line, where each combination of $CX, X$ and Hadamard gate ($H$) gates creates a singlet. Next, to the right of the red line, the 3-qubit unitary operation $\hat{\mathcal{U}}$ from Eq. (7) is effected, where every third qubit $q_{3k}$ is an ancilla. To recover the non-unitary spin-1 projection $\hat{\mathcal{P}}$ from Eq. (6), post-selection "$\langle0|$" operations are performed on the ancilla qubits, as described by Eq. (10). The OBC AKLT state shown in FIG. 1(a) is obtained through measurements and post-selections on the physical qubits. With the circuit geometry given in FIG. 1(b), the $CX$ gates between $q_2, q_4$ and $q_5, q_7$ act between nearest neighbor qubits when embedded in a quantum processor (also see Fig. 4).

$$\begin{aligned}|\psi\rangle_{\text{AKLT}}&=\left(\bigotimes_{k=0}^{L-1}\langle\uparrow|_{3k}\right)\left[\hat{\mathcal{U}}\left(0,1,2\right)\otimes\hat{\mathcal{U}}\left(3,4,5\right)\otimes\cdots|\psi\rangle_0\right]\\&=\left(\bigotimes_{k=0}^{L-1}\langle\uparrow|_{3k}\right)\left[\prod_{j=0}^{L-1}\hat{\mathcal{U}}\left(3j,3j+1,3j+2\right)|\psi\rangle_0\right]\\&=\prod_{j=0}^{L-1}\hat{\mathcal{P}}_j\,|\phi\rangle_0\end{aligned}$$

$$(14)$$

where $|\psi\rangle_0 = \prod_{k=0}^{L-1} |\uparrow\rangle_{3k} \otimes |\phi\rangle_0$ is the product state of all the ancilla qubits in spin up $(\prod_{k=0}^{L-1} |\uparrow\rangle_{3k})$, and singlet pairs $(|\phi\rangle_0)$ generated from the first step. For each $j$, $\hat{\mathcal{P}}_j$ is the same projection operator as $\hat{\mathcal{P}}$ in Eq. (6). $\langle\uparrow|_{3k}$ $(k = 0, 1, 2, \cdots, L-1)$ represents the projection i.e. postselection of the ancilla qubits onto spin up. We use the notation $\hat{\mathcal{U}}(3j, 3j+1, 3j+2)$ to indicate that the unitary operator $\hat{\mathcal{U}}$ acts on qubits $3j, 3j+1$ and $3j+2$. The last line from the above Eq. (14) is basically a product version of Eq. (11). From there, by simply following the same way as how the AKLT state is constructed by the projection operator in the MPS formalism in Refs. [65, 67], one could obtain the exact expression of the AKLT state for both OBC and PBC in Eq. (3) and (2), respectively.

## C.  Variational circuit recompilation for the three-qubit unitary operator

In general, unitary operators on a quantum circuit are transpiled in terms of the basis gates and the device geometry of the physical quantum processor. On the IBM Q device, two-qubit gates such as $CX$ gates and SWAP gates, incur non-negligible error, and a practical challenge is to reduce the number of such two-qubit gates as far as possible. Explicitly, the IBM Q device which we use have $CX$ gate error rate from 0.6% to 5% (see FIG. 4), and accordingly, any circuit containing more than a few hundred CX gates is not ideal for robust simulation on such a quantum device.

In our work of realizing the AKLT state, a crucial step is implementing the three-qubit operation in Eq. (7) on a quantum circuit. A straightforward approach has so far been the isometry decomposition (see Appendix A), but that involves a large number of two-qubit $CX$ gates as well as single-qubit gates [72]. For $\hat{\mathcal{U}}$, the `transpile` function [73] from Qiskit requires at least 24 $CX$ gates.

Moreover, of these $CX$ gates, 8 are not nearest-neighbor, and this will further require SWAP gates after being transpiled to the quantum hardware if these three qubits are aligned as a linear chain.

On today's NISQ-era quantum computers, we are aware that the Variational Quantum Algorithms (VQAs) are effective methods for the current NISQ-era device [74, 75] with reduced number of gates. In VQAs, parameterized circuits are first obtained on a classical computer by an optimization algorithm, and then these circuits with optimized parameters are executed on the quantum computer. As such, we consider a variational approach known as the circuit recompilation [17, 18, 58, 59, 76], which has been shown to give promising approximations to the original unitary whilst having much shallower circuit depths, and with fewer $CX$ and single-qubit gates compared to the default isometry decomposition. This will result in significantly lower aggregate gate error on current NISQ-era quantum processors.

To conduct the circuit recompilation, we consider the

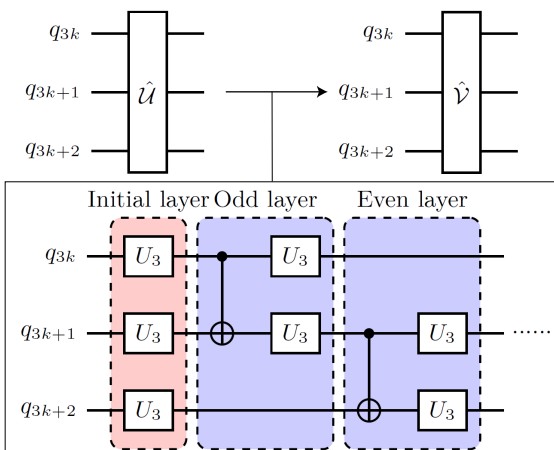

FIG. 3. **Variational circuit recompilation of the three-qubit unitary operator $\hat{\mathcal{U}}$.** The unitary operator $\hat{\mathcal{U}}$ in FIG. 2 is replaced by a ansatz circuit $\hat{\mathcal{V}}$ consisting of an initial layer of three single-qubit $U_3$ gates (pink) followed by $n_l$ layers, each containing one $CX$ gate and two $U_3$ gates (purple). Show here are $n_l = 2$ layers. The optimized parameters for every $U_3(\phi, \theta, \lambda)$ gate are obtained by training a tensor network on a classical computer [17, 58, 59].

ansatz shown in FIG. 3, where the original 3-qubit unitary operator $\hat{\mathcal{U}}$ is substituted with a variational circuit $\hat{\mathcal{V}}$ consisting of an initial layer of single-qubit $U_3$ rotations (pink block), followed by $n_l$ layers, each consisting of two $U_3$ gates and a CX gate (purple blocks). The CX gate acts between the "middle" qubit and either of the other two qubits, depending on whether the layer index is odd or even. This recompiled circuit consists of $9 + 6n_l$ variational parameters, with each 3D rotation gate $U_3(\phi, \theta, \lambda)$ parametrized by three rotational parameters $\phi, \theta$ and $\lambda$ that are optimized through a limited memory Broyden-Fletcher-Goldfarb-Shanno algorithm with box constraints (L-BFGSB) [17, 77, 78]. To avoid being trapped in the local minima, we use a basin-hopping method [79–82], where small perturbations are added to each optimization round followed by local minimization for each step, and the search is from $n_l = 5$ layers to $n_l = 9$ layers. The loss function is constructed as follows: for both target unitary operator $\hat{\mathcal{U}}$ and the ansatz $\hat{\mathcal{V}}$ operator, each is first reshaped to a rank-$2M$ $(M = 3L)$ tensor where each index has a dimension of two. By contracting these two tensors $\hat{\mathcal{V}}$ and $\hat{\mathcal{U}}$ as a scalar, and after normalizing and negating it, we have

$$f\left(\hat{\mathcal{U}}, \hat{\mathcal{V}}\right) = 1 - \frac{1}{2^M} \sum_{j_1, \cdots, j_M} \sum_{i_1, \cdots, i_M} \hat{\mathcal{U}}_{i_1, \cdots, i_M}^{j_1, \cdots, j_M} \hat{\mathcal{V}}_{j_1, \cdots, j_M}^{i_1, \cdots, i_M},$$
(15)

where $\hat{\mathcal{U}}_{i_1, \cdots, i_M}^{j_1, \cdots, j_M}$ and $\hat{\mathcal{V}}_{j_1, \cdots, j_M}^{i_1, \cdots, i_M}$ are the rank-$2M$ tensors reshaped from their corresponding operator. We then obtain the loss function $f\left(\hat{\mathcal{U}}, \hat{\mathcal{V}}\right)$ for the optimization [83]. See Appendix B for details of the optimization process.

Since the loss function from Eq. (15) itself is constructed in a way that is independent of any initial state, the validity of the recompiled circuit can be simply characterized by computing the fidelity between a random state $|\beta\rangle$ acted by the original target operator (which is $\hat{\mathcal{U}}$ here), and the same state acted by the recompiled operator $\hat{\mathcal{V}}$:

$$\mathcal{F}(\hat{\mathcal{V}}|\beta\rangle, \hat{\mathcal{U}}|\beta\rangle) = \langle\beta|\hat{\mathcal{V}}^\dagger\hat{\mathcal{U}}|\beta\rangle. \qquad (16)$$

In our work, we achieved very high fidelity $\mathcal{F} > 99.99\%$ of the recompiled $\hat{\mathcal{V}}$ gates typically with just $n_l = 8$ variational layers. Also, compared with other recent proposals of implementing the non-unitary operator using imaginary time evolution [31, 84], our approach consists of only one step of unitary evolution from a trival singlet product state plus ancilla qubits in spin-up, followed by a measurement-based post-selection. The corresponding variational circuit renders a sufficiently shallow circuit such that the outcomes are robust against the quantum gate infidelity on IBM Q. More details are given below FIG. 9 of the appendix.

## D. Implementation layout on IBM quantum processors

To prepare the AKLT state on actual quantum processors, we need to embed the quantum circuits from Figs. 2 and 3 onto suitable device layouts. To maximize efficiency and minimize gate errors, it is highly preferable that the logical structure of the qubit couplings [Fig. 2(b)] conforms as closely as possible to the actual physical couplings within the quantum processor (if not, more distant couplings can still be effected by "stacking" $CX$ gates [18], but doing so introduces greater gate errors.). In particular, since we require one ancilla qubit for every two qubits in the logical spin-1/2 chain, we should ideally have an uninterrupted chain of $2L$ qubits such that every even (or odd) qubit is connected to an additional ancilla qubit.

In FIG. 4, we show how we embedded AKLT states of different sizes $L = 2$ [FIG. 4(a)], $L = 3$ [FIG. 4(b)], $L = 4$ [FIG. 4(c)] and $L = 5$ [FIG. 4(d)] on the IBM quantum processor (Throughout this work, we use "$ibmq\_montreal$"). These configurations are also selected because they are susceptible to the lowest amounts of gate errors, as according to calibration data. The physical (ancilla) qubits are highlighted using red (green) squares, and the grey arrows indicates the direction of ascending qubit labels (from $q_0$ to $q_{3L-1}$).

## E. State measurement and post-selection

After the state preparation, we need to compellingly measure the putatively prepared state to check whether

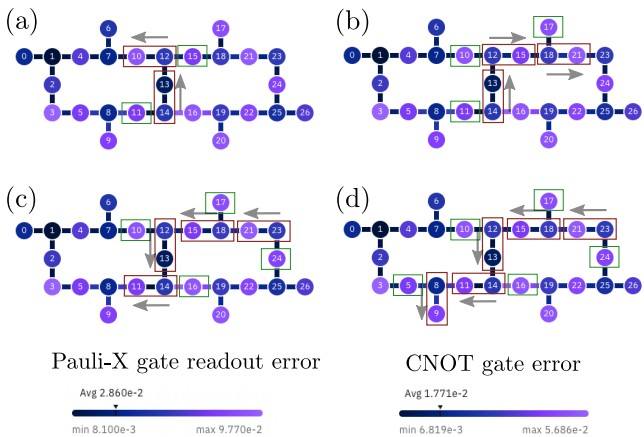

FIG. 4. **Layouts of the utilized qubits on our IBM quantum processor** (a) - (d) shows the selected qubits in the *ibmq_montreal* quantum computer preparation of sized $L = 2, 3, 4$ and 5 AKLT states, respectively. The spin-1/2 chain (ancilla) qubits are highlighted by red (green) solid squares. The grey arrow points the direction of each qubit chain from the beginning. For different system sizes $L = 2, 3, 4$ and 5, the qubits chosen on *ibmq_montreal* are: (a) $[11, 14, 13, 15, 12, 10]$, (b) $[11, 14, 13, 10, 12, 15, 17, 18, 21]$, (c) $[24, 23, 21, 17, 18, 15, 10, 12, 13, 16, 14, 11]$ and (d) $[24, 23, 21, 17, 18, 15, 10, 12, 13, 16, 14, 11, 5, 8, 9]$, with gray arrows indicating the directions of ascending qubit labels. The error for single-qubit Pauli-$X$ gates and $CX$ gates are also shown.

the AKLT state was indeed realized. The IBM quantum computer only allows for measurements that outputs whether a qubit is spin up or down - by repeating a large number of "shots" or "runs", the expectation value of $\langle\tau_z\rangle = \langle q|\tau_z|q\rangle$ of a qubit $q$ can be measured [85].

Following the quantum circuit being executed on the IBM quantum processor over a large number of shots, we perform post-selection not just to project out the ancilla qubit for effectively non-unitary evolution, but also to mitigate the effect of noise on the data so as to enhance the signal-to-noise ratio. To implement $\hat{\mathcal{P}}$ from Sec. III A via post-selection, the instances where the ancilla qubits are all measured to be spin up ('$|\uparrow\rangle$') are recorded, and those with at least one spin down ('$|\downarrow\rangle$') are discarded. Of those instances that are post-selected thus far, we can perform another round of post-selection to eliminate spurious instances compromised by noise. We only keep instances whose bit strings conserve the total spin up numbers i.e. with the same number of '$|\uparrow\rangle$' in the context of IBM Q data of counts. This is because the projection operator, sometimes also called the symmetrization operator [66] from Eq. (6), does not change the total spin number, and therefore the state itself after the application of the unitary operator should have the same number of spin up ('$|\uparrow\rangle$') with the initial product state of singlets. After that, the probability amplitude for each qubit of the AKLT state in the spin-1/2 basis can be calculated and compared with the exact values from

the MPS, which is discussed in the following section (see also Appendix. C).

## IV.  MEASUREMENT RESULTS

In this section, we present and evaluate the performance of our algorithm for preparing the OBC AKLT state on a quantum processor. We first introduce the Hellinger fidelity for quantifying how closely the prepared AKLT state agrees with the exact simulated AKLT state in Sec. IV A. Next, we present measurement results of the AKLT state with different sizes in Sec. IV B, and briefly discuss some broader implications.

### A.  State fidelity

To evaluate the validity of our state-preparation algorithm of FIG. 3, we use the Hellinger fidelity [86]. This quantity can directly estimate the similarity between two probability distributions, which is suitable for the sampling statistics nature of IBM Q data [87]. Most recently, this quantity has been used to characterize the performance of Dicke state preparation on the IBM Q system [88], as well as to investigate the quantum circuit reproducibility [89]. In analogy to classical probabilities, for any two states represented in the spin-$1/2$ basis as

$$|R\rangle = \sum_i r_i |\sigma_i\rangle, |S\rangle = \sum_i s_i |\sigma_i\rangle, \qquad (17)$$

the Hellinger fidelity is defined as

$$F(|R\rangle, |S\rangle) = \left[\sum_i \sqrt{|r_i|^2 |s_i|^2}\right]^2. \qquad (18)$$

If $|R\rangle$ and $|S\rangle$ were identical, all the coefficients $\sqrt{|r_i|^2|s_i|^2} = |r_i|^2$ would sum to unity; otherwise, the departure of $F$ from unity signifies the lack of perfect agreement between $|R\rangle$ and $|S\rangle$. Here, we take $|R\rangle$ and $|S\rangle$ as the AKLT state prepared on the quantum processor and the exact AKLT state simulated using the local noiseless Qiskit `aer_simulator` backends on the Qiskit Terra circuits respectively. In other words, $|r_i|^2$ is the probability distribution obtained from measuring the physical quantum circuit and $|s_i|^2$ represent the exact AKLT state probability distribution.

### B.  Verification of the AKLT state

In FIG. 5, we present very good agreement between the OBC AKLT states prepared on the IBM quantum computer (with and without error mitigation) with the results from the noiseless local simulator. Here, the signatures of the AKLT state characterization are the probability amplitudes of each basis component for $L = 2$

and $L = 3$, which are obtained via post-selection of the raw measurement output from the IBM quantum computer[90]. We observe qualitative agreement with the results from the noiseless `aer_simulator` for $L = 2$ and 3 in Fig. 5(a) - (c). As for $L = 5$, which is only presented in Fig. 6, the numerical readout error mitigation is extremely costly, i.e. one needs $2^{15}$ circuits to construct the calibration matrix, which is beyond the maximal number of circuits each IBM Q device could host per submission. Therefore, the readout error mitigation is performed using the `mthree` package [91] without explicitly constructing a calibration matrix (see details of error mitigation in Appendix. D). We execute all the circuits, including those for readout error mitigations at the same time, so as to reduce the effect of stochastic noise on the device as much as possible. Therefore, for each circuit for $L$, the result is calculated and averaged over 91 repeated executions of the same circuit on *ibmq_montreal*. This is the maximum allowable number for which each circuit for $L$ can be repeated on this device. The error bars represent the standard error.

As mentioned above, the number of $CX$ gates which grows linearly with the system size $L$ in the recompiled quantum circuit mainly contributes to the error in the simulation on the IBM Q device, and FIG. 5 illustrates such effect clearly.According to the case with $L = 2$ in FIG. 5(a), we find that our results are very close to the exact values for all component basis states. However, for $L = 3$ in FIG. 5(b) and (c), only the value for the basis $\langle\uparrow\uparrow\downarrow\downarrow\uparrow\uparrow\rangle$ is close to the exact one, while most of the others exhibit some visible deviations from the exact values. Overall, the averaged Hellinger fidelity $\overline{F}(|\psi\rangle_L, |\Phi\rangle)$ starts to drop dramatically after $L = 3$ where $|\Phi\rangle$ is the state obtained from noiseless Qiskit `aer_simulator`. For both $L = 2$ and $L = 3$ cases, compared with the unmitigated results, the error mitigation does not show an obvious improvement. Since our error mitigation method only focuses on the readout error, the effect of the error mitigation is much more significant for larger systems where more qubit measurements are required.

To check how fast the fidelity decreases when the system size increases, or whether a high fidelity can be sustained, we study the averaged Hellinger fidelity $\overline{F}(|\psi\rangle_L, |\Phi\rangle)$ versus $L$ in FIG. 6. As also apparent in FIG. 5, we observed that for larger systems, the state fidelity decreases quickly for the default transpiled AKLT state, indicating a poor state preparation. This is because the total number of $CX$ gates increases linearly with respect to the system size $L$, and therefore $CX$ gate errors quickly become the most dominant source of gate fidelity errors in the quantum processor. In that case, our approach for the readout-error mitigation will be less effective when $L$ increases. Moreover, compared to the fidelity for different $L$ using the default `transpile` function from Qiskit, the advantage of our variational approach is more obvious for larger systems, although the fidelities for smaller systems with $L = 2$ are almost the same. Our findings indicate that the variational

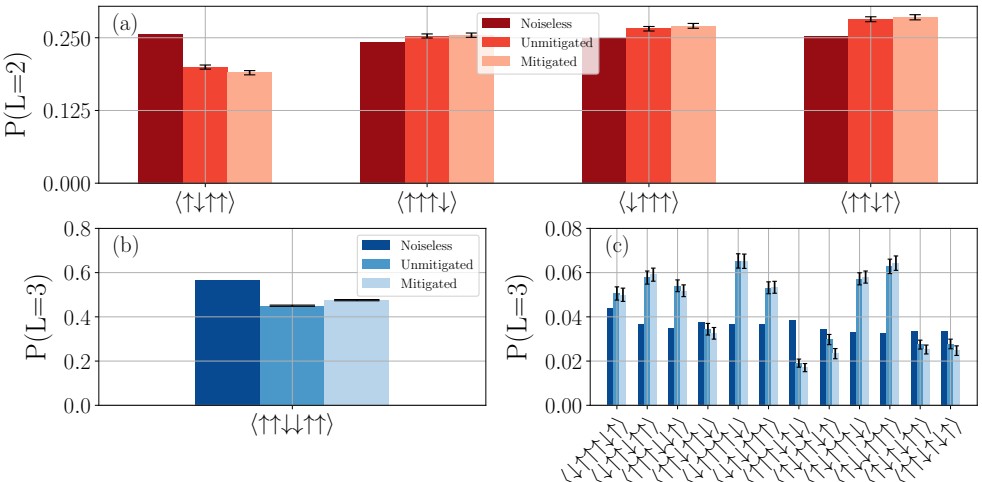

FIG. 5. **Characterization of AKLT state preparation with open boundary conditions.** (a) Comparison of the probability amplitudes of the $L = 2$ AKLT state components in the spin-1/2 basis. (b) and (c) comparison of the probability amplitudes of the $L = 3$ AKLT state components in the spin-1/2 basis. In panels (a), (b) and (c), from darker to lighter color, the bar indicates the ideal local noiseless simulation results from `aer_simulator`, the unmitigated and mitigated results from the real devices. For all panels, the mitigated and unmitigated results are obtained from *ibmq_montreal*, and error bars are calculated based on 91 repeated executions. A maximum of 32000 shots is used for each execution, and therefore the effective shots for each circuit is $91 \times 32000 = 2912000$.

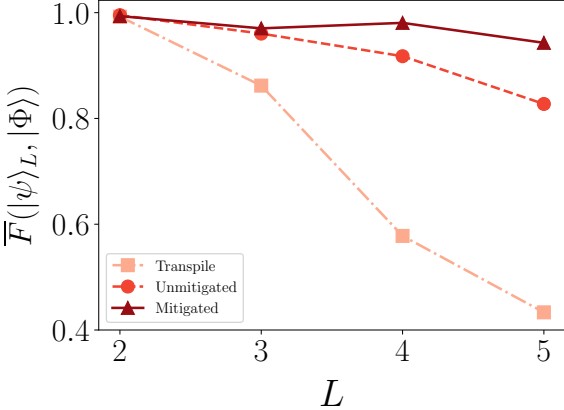

FIG. 6. **Averaged Hellinger fidelity $\overline{F}(|\psi\rangle_L, |\Phi\rangle)$ as a function of system size $L$.** From darker to lighter color, the line represents the mitigated results using variational parametrized circuits (solid line with triangles), the unmitigated results using variational parameterized circuits (dashed line with circless), and those results obtained using default `transpile` function from Qiskit (dot-dashed line with squares). All results were obtained from *ibmq_montreal* averaged over 91 repeats of executions. $|\Phi\rangle$ is the state obtained from noiseless Qiskit `aer_simulator`. Notably, the recompiled variational circuit ansatz, whether with or without error mitigation, significantly preserves the fidelity as $L$ increases.

parametrized (recompiled) circuit with fewer $CX$ gates is capable of maintaining a higher state fidelity and more accurate probability amplitudes for larger system sizes (see also Appendix. A). Furthermore, according to the

result under the readout error mitigation in FIG. 6, although the improvement in the fidelity value is modest, the effect of error mitigation is more significant for larger system sizes. This is because a larger system size requires more measurements during the state's preparation. As a result, once the gate fidelity on quantum processors improves in the near future, our algorithm will achieve significantly higher fidelities state preparation even at larger $L$.

Our algorithm can also be naturally extended to perform the preparation of the AKLT state with PBCs, although, for the current stage, this is restricted by the geometry of IBM Q devices, as a specific qubit ring is needed such that both edge spins are connected. In Appendix. C 2, we show the results for a local classical noiseless simulation of the preparation of the AKLT state with PBC without noise, which shows good agreement with the values computed from the MPS representation [92]. Also, as our circuit is notably shallower than the one generated using the default `transpile` function from Qiskit, it has the capacity for further operations on the obtained AKLT state, e.g. performing quench dynamics of the state, or using the state for the calculation of observables interested by post selecting all the ancilla qubits to be in $|\uparrow\rangle$.

## V. CONCLUSION

In conclusion, we presented an efficient algorithm to prepare the AKLT state on an IBM quantum computer. By using an additional ancilla qubit, we are able to embed the non-unitary projection operator into a unitary

operator acting on three qubits. Through the variational recompilation of the operator, such a three-qubit operation is then entirely transformed into a parametrized circuit with a reduced circuit depth. This approach is non-deterministic, and therefore we show the state can be obtained by evolving a trivial initial product state of singlets plus ancilla qubits in spin-up using this parametrized circuit, and then by post-selection and spin number conservation. The simulations on the noisy IBM quantum processor show that the state fidelity is higher for those with smaller system sizes, and due to the aggregate $CX$ gate error when the system size is larger, the lower fidelity indicates the poorer state preparation. Accordingly, the effect of readout-error mitigation on the state fidelity is more obvious in cases of larger system sizes. In terms of the variational recompilation of the quantum circuit, our approach provides an efficient way to both prepare AKLT state with fewer $CX$ and single qubit gates on NISQ-era quantum processors, and for subsequent operations using the state preparation. Also, the evolution from the initial state only has one step, and it does not require any mid-circuit measurement.

In the future, this work will inspire similar algorithms for higher-dimensional AKLT states [34, 35, 45, 46, 93, 94], or other types of VBS states which are of great interest to the condensed matter physics community. More careful studies should also be performed to increase the state fidelity for larger system sizes with suitable error mitigation techniques and adequate numerical resources. With appropriate redefinitions of the basis states, the exponentially large Hilbert space of even modestly sized quantum processors can be used to demonstrate the physics in large multi-dimensional lattices [95–98], as already demonstrated in Ref. [18]. Another possible direction is to come up with quantum algorithms for the computation of the relevant quantities from the AKLT state which is already prepared on a quantum computer.

This is of great importance to the application of quantum computers on many-body physics, but is still at its infancy in current literature.

We also note that non-unitary operators also describe the time evolution of effectively non-Hermitian systems. As such, with some modifications, our algorithm can be adapted to simulate non-Hermitian many-body phenomena as well as entanglement dynamics on the quantum computer [99–111]. As a concrete case in point, the phase transition associated with $\mathcal{PT}$ symmetry breaking is generally induced by gain or loss, which can be realized by coupling system qubits to an ancilla qubit analogously to our approach [112–114]. Moreover, in terms of realizing loss as dissipation, one can simulate quantum systems with dissipative boundaries [115–117]. In addition, under appropriate generalizations, non-unitary dynamics can be directly embedded in a quantum circuit, which would facilitate the simulation of dissipative quantum dynamics on a quantum computer. [118–123].

## ACKNOWLEDGMENTS

T. C. thanks Tim Byrnes for fruitful discussions. T. C. and R. S. thank Truman Ng for discussions on the quantum simulation implementation on IBM Quantum services. T. C. and B. Y. acknowledges support from the Singapore National Research Foundation (NRF) under NRF fellowship award NRF-NRFF12-2020-0005. C. H. L. acknowledges support from the Singapore's NRF Quantum engineering grant NRF2021-QEP2-02-P09. We acknowledge the use of IBM Quantum services for this work. The views expressed are those of the authors, and do not reflect the official policy or position of IBM or the IBM Quantum team. The diagrams of quantum circuits in this work were partially produced using Quantikz [124].

T. C. and R. S. contributed equally to this work.

---

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

## Appendix A: Transformation to an unitary operator

To realize the non-unitary projection $\hat{\mathcal{P}}$ in Eq. (6) from the main text, we consider embedding it into a three-qubit operation $U$ via the following ansatz [29]

$$\hat{\mathcal{U}} = \left[ \begin{array}{c|c} \hat{\mathcal{P}} & \hat{\mathcal{A}} \\ \hline \sqrt{\hat{I} - \hat{\mathcal{P}}^{\dagger}\hat{\mathcal{P}}} & \hat{\mathcal{B}} \end{array} \right] \tag{A1}$$

with identity matrix $\hat{I}$. Thus, $\hat{\mathcal{U}}$ can be solved by the following $\hat{\mathcal{U}}'$ ansatz

$$\hat{\mathcal{U}}' = \left[ \begin{array}{c|c} \hat{\mathcal{P}} & \hat{I} \\ \hline \sqrt{\hat{I} - \hat{\mathcal{P}}^{\dagger}\hat{\mathcal{P}}} & \hat{I} \end{array} \right] = \hat{\mathcal{U}}\hat{\mathcal{R}}. \tag{A2}$$

Here, $\hat{\mathcal{R}}'$ is computed by the QR decomposition of $\hat{\mathcal{U}}'$, which accordingly gives the solution of $\hat{\mathcal{U}}$ in Eq. (7). The target state from the operation $\hat{\mathcal{U}}$ is obtained by the post-selection given in Eq. (10) of the main text.

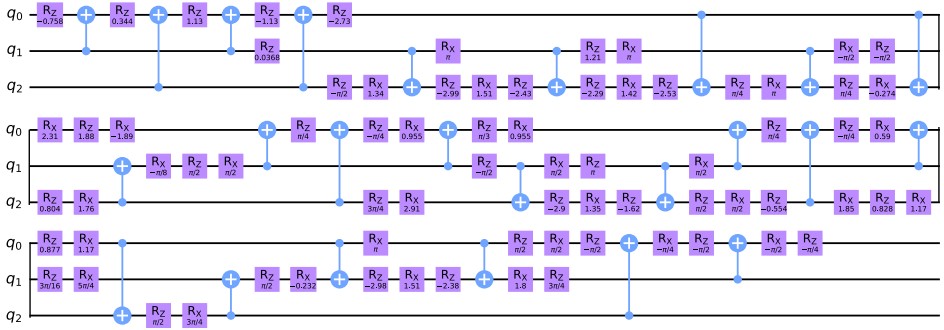

FIG. 7. Numerical decomposition of the three-qubit operation in Eq. (7). The decomposition requires 24 CX-gates which are more than the circuit under 8 layers from the variational approach in FIG. 3. The circuit diagram is generated using Qiskit.

As mentioned in the main text, the three-qubit operation in Eq. (7) can also be realized with the isometry decomposition in FIG. 7 which requires 24 $CX$ gates. However, the circuit based on our variational approach described in FIG.3 requires much fewer gates: 8 $CX$ gates for a total of 8 layers. The corresponding comparison of the result is shown in FIG. 8. Here, the basis states are those component basis of the corresponding AKLT state ($L = 2$ and $L = 4$) represented in the spin-1/2 basis similar to those shown in Fig. 5 [127]. For smaller system such as $L = 2$, the probability amplitude for both approaches are almost equally close to the exact values [FIG. 8(a)]. However, the effective number of shots for our approach is almost two times more than the default `transpile` approach [FIG. 8(c)]. Once the system size goes larger ($L = 4$), for certain basis states, especially for those with larger probability amplitude values, our approach shows that they are closer to the exact values than the default `transpile` approach [FIG. 8(b)], and more effective number of shots [FIG. 8(d)] as well. As a result, it renders better state fidelity, as already discussed in the main text. All simulations were executed on *ibm_montreal*. Our approach also indicates that since the circuit is shallower and there are more effective number of shots, it is capable of further operations once the AKLT state is prepared.

## Appendix B: Variational circuit optimization

In FIG. 9, we illustrate the performance of the variational circuit parameterization by showing both the averaged circuit fidelity $\overline{\mathcal{F}}$ and the maximum fidelity $\mathcal{F}_{\max}$ with respect to different number of layers $n_l$, as well as different number of iterations of the optimizations. The averaged $\overline{\mathcal{F}}$ and the error bars are computed over 20 rounds of repeats of the complete optimization. It is found that an iteration number of 600 can already render a parameterized circuit fidelity closer to 1, as shown in FIG. 9. With layer $n_l = 8$, the optimization could result in an averaged circuit fidelity $\overline{\mathcal{F}}$ close to 1 with fixed number of iterations and basin hoppings [FIG. 9(a)]. For $n_l = 5, 6, 7$, they fail to achieve a fidelity larger than the case of $n_l = 8$ within all 20 rounds of repeat. When $n_l = 9$, though $\overline{\mathcal{F}}$ is not better than the case of $n_l = 8$ due to the other parameters chosen, the largest $\mathcal{F}_{\max}$ for $n_l = 9$ still gives a fidelity larger than 99.99% [FIG. 9(b)]. Therefore, throughout this work, we choose $n_l = 8$ as the least number of $CX$ and single-qubit gates to

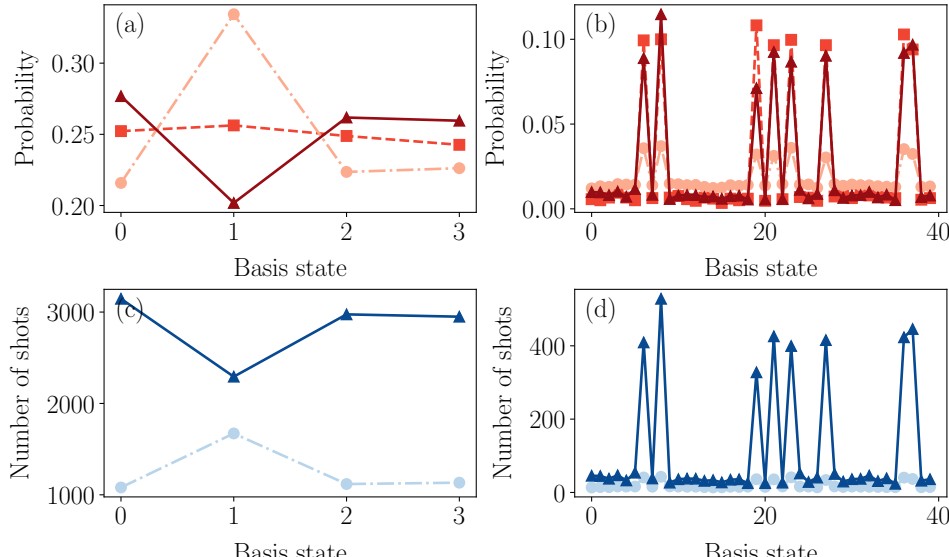

FIG. 8. Probability amplitude for (a) $L = 2$ and (b) $L = 4$ AKLT basis states. From lighter to darker color, the curve for each panel represents the results obtained using the default `transpile` approach (dot-dashed line), the noiseless `aer_simulator` (dashed line), and the variational parameterized circuit (solid line); The effective number of shots for (c) $L = 2$ and (d) $L = 4$ AKLT basis states. From lighter to darker color, the curve for each panel represents the results obtained using the default `transpile` approach (dot-dashed line), and the variational parameterized circuit (solid line). For all panels, the $x$ axis stands for basis states (we omit the expression for each basis state for purpose of simple presentation), and the total shots for each execution is 32000, which is the maximum number of shots for *ibm_montreal*. The basis states are those component basis of the AKLT state represented in the spin-1/2 basis.

minimize the gate fidelity error. Comparing our outcomes with the numerical decomposition of $\hat{\mathcal{U}}$ using the default `transpile` function from Qiskit, our approach results in fewer $CX$ gates for each unitary operator, and the number of the $CX$ gates scales linearly with the size of the AKLT state. Therefore, our finding shows that we are able to realize the same unitary operator on a quantum circuit with much shallower circuit depth.

## Appendix C: Explicit exact forms of the AKLT state for $L = 2$ and $L = 3$

To characterize the validity of our prepared AKLT state, we check whether the state components in spin-1/2 basis are the same as those from the exact MPS representations, and how close their corresponding probability amplitudes obtained from the real IBM Q device on are to the exact value. We remark that since the $A^\sigma$ matrices from Eq. (4) only normalize the whole state in the thermodynamic limit [66, 70], we calculate the probability amplitude distribution from the normalized coefficients, and compare them with the results from IBM Q after performing the post-selection. In the following context, for simplicity, we calculate the exact values for both OBC and PBC at $L = 2$ as an example.

### 1. Open boundary conditions

For simplicity, we first show a $L = 2$ AKLT state with OBC with details. The non-trivial basis states calculated from the exact MPS expression are:

$$|\psi\rangle_{L=2}^{\text{OBC}} = \alpha_1 |O\rangle |+\rangle + \alpha_2 |+\rangle |O\rangle \tag{C1}$$

and

$$\alpha_1 = {b_A^l}^T A^O A^+ b_A^r = -\frac{\sqrt{2}}{3}$$
$$\alpha_2 = {b_A^l}^T A^+ A^O b_A^r = \frac{\sqrt{2}}{3} \tag{C2}$$

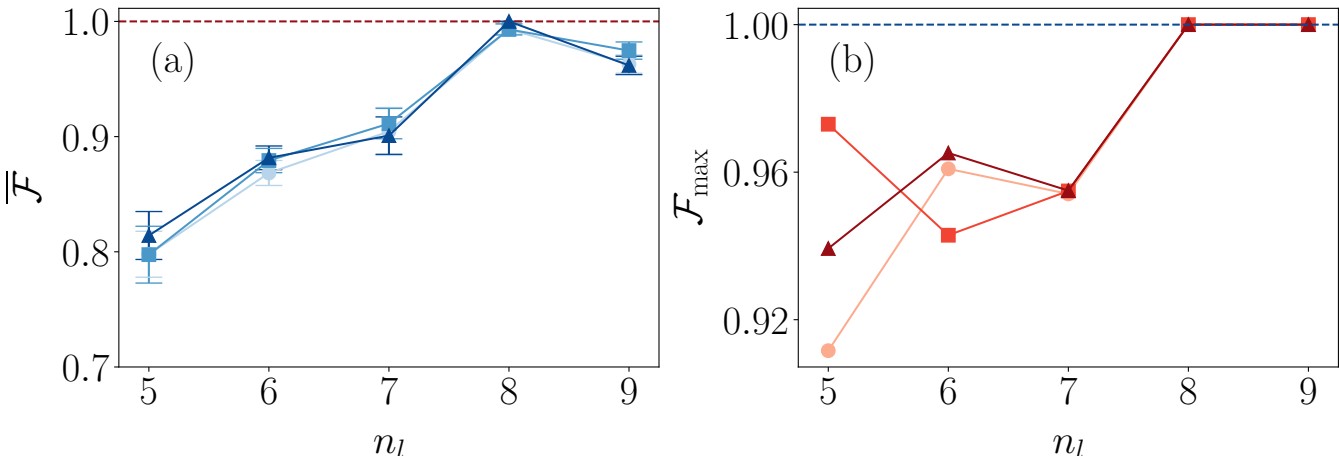

FIG. 9. Effect of number of variational ansatz labels $n_l$ on the validity of circuit recompilation: (a) Averaged circuit recompilation fidelity $\bar{\mathcal{F}}$ as a function of the number of recompiled circuit layers $n_l$. For all the curves, the darker blue color represents the larger number of iterations: 600 (circle), 700 (square) and 800 (triangle). The error bar of each data is obtained by calculating the standard error from 20 rounds of optimizations. (b) Maximum circuit recompilation fidelity $\mathcal{F}_{\max}$ as a function of the number of recompiled circuit layers $n_l$. For all the curves, the darker red color represents the larger number of iterations: 600 (circle), 700 (square) and 800 (triangle). For all panels, the number of basin hopping is fixed at 20. The dashed line indicates the value of fidelity equal to 1.

where $b_A^{l\ T} = \begin{pmatrix} 1 & 0 \end{pmatrix}^T$, $b_A^r = \begin{pmatrix} 0 & 1 \end{pmatrix}^T$, and $A^O = -\sqrt{\frac{1}{3}}\tau^z$, $A^+ = +\sqrt{\frac{2}{3}}\tau^+$ which are the same expressions from Eq. (4) and (3). As the state in Eq. (C1) is unnormalized, we first expand each in the spin-1/2 basis by substituting $|+\rangle = |\uparrow\uparrow\rangle$, and $|O\rangle = 1/\sqrt{2}\,(|\uparrow\downarrow\rangle + |\downarrow\uparrow\rangle)$ into Eq. (C1):

$$|\psi\rangle_{L=2}^{\mathrm{OBC}} = \frac{\alpha_1}{\sqrt{2}}\,(|\uparrow\downarrow\rangle + |\downarrow\uparrow\rangle)\,|\uparrow\uparrow\rangle + \frac{\alpha_2}{\sqrt{2}}\,|\uparrow\uparrow\rangle\,(|\uparrow\downarrow\rangle + |\downarrow\uparrow\rangle) \tag{C3}$$
$$= \frac{\alpha_1}{\sqrt{2}}\,|\uparrow\downarrow\uparrow\uparrow\rangle + \frac{\alpha_1}{\sqrt{2}}\,|\downarrow\uparrow\uparrow\uparrow\rangle + \frac{\alpha_2}{\sqrt{2}}\,|\uparrow\uparrow\uparrow\downarrow\rangle + \frac{\alpha_2}{\sqrt{2}}\,|\uparrow\uparrow\downarrow\uparrow\rangle$$

As stated above, this state is not normalized. We introduce a normalization factor $\mathcal{N} = \sqrt{2\left(\alpha_1/\sqrt{2}\right)^2 + 2\left(\alpha_2/\sqrt{2}\right)^2}$ which exactly corresponds to the post-selection procedure in Sec. III A of the main text, and compute the normalized coefficient for the wavefunction as

$$\tilde{\alpha}_1 = \frac{1}{\mathcal{N}}\frac{\alpha_1}{\sqrt{2}} = -\frac{1}{2} \tag{C4}$$
$$\tilde{\alpha}_2 = \frac{1}{\mathcal{N}}\frac{\alpha_2}{\sqrt{2}} = \frac{1}{2}$$

and the state itself becomes

$$\left|\tilde{\psi}\right\rangle_{L=2}^{\mathrm{OBC}} = \tilde{\alpha}_1\,|\uparrow\downarrow\uparrow\uparrow\rangle + \tilde{\alpha}_1\,|\downarrow\uparrow\uparrow\uparrow\rangle + \tilde{\alpha}_2\,|\uparrow\uparrow\uparrow\downarrow\rangle + \tilde{\alpha}_2\,|\uparrow\uparrow\downarrow\uparrow\rangle . \tag{C5}$$

Therefore, the normalized probability amplitude for each component basis is then

$$P\left[|\uparrow\downarrow\uparrow\uparrow\rangle\right] = \frac{1}{4}, P\left[|\downarrow\uparrow\uparrow\uparrow\rangle\right] = \frac{1}{4} \tag{C6}$$
$$P\left[|\uparrow\uparrow\uparrow\downarrow\rangle\right] = \frac{1}{4}, P\left[|\uparrow\uparrow\downarrow\uparrow\rangle\right] = \frac{1}{4}$$

For $L = 3$, by following the same procedure as decribed above, we can obtain

$$|\psi\rangle_{L=3}^{\mathrm{OBC}} = \alpha_1\,|+\rangle\,|-\rangle\,|+\rangle + \alpha_2\,(|+\rangle\,|O\rangle\,|O\rangle - |O\rangle\,|+\rangle\,|O\rangle + |O\rangle\,|O\rangle\,|+\rangle) \tag{C7}$$

where $\alpha_1 = -2\sqrt{6}/9$ and $\alpha_2 = \sqrt{6}/9$. After inserting the expression of $|O\rangle$ and the normalization, the state itself becomes

$$\left|\tilde{\psi}\right\rangle_{L=3}^{\text{OBC}} = \tilde{\alpha}_1 \left|\uparrow\uparrow\downarrow\downarrow\uparrow\uparrow\right\rangle + \tilde{\alpha}_2 \left(\left|\uparrow\uparrow\uparrow\downarrow\uparrow\downarrow\right\rangle + \left|\uparrow\uparrow\downarrow\uparrow\uparrow\downarrow\right\rangle + \left|\uparrow\uparrow\uparrow\downarrow\downarrow\uparrow\right\rangle + \left|\uparrow\uparrow\downarrow\uparrow\downarrow\uparrow\right\rangle\right) \tag{C8}$$
$$- \tilde{\alpha}_2 \left(\left|\uparrow\downarrow\uparrow\uparrow\uparrow\downarrow\right\rangle + \left|\downarrow\uparrow\uparrow\uparrow\uparrow\downarrow\right\rangle + \left|\uparrow\downarrow\uparrow\uparrow\downarrow\uparrow\right\rangle + \left|\downarrow\uparrow\uparrow\uparrow\downarrow\uparrow\right\rangle\right) + \tilde{\alpha}_2 \left(\left|\uparrow\downarrow\uparrow\downarrow\uparrow\uparrow\right\rangle + \left|\downarrow\uparrow\uparrow\downarrow\uparrow\uparrow\right\rangle + \left|\uparrow\downarrow\downarrow\uparrow\uparrow\uparrow\right\rangle + \left|\downarrow\uparrow\downarrow\uparrow\uparrow\uparrow\right\rangle\right)$$

and therefore the normalized probability amplitude for each component basis is

$$P\left[\left|\uparrow\uparrow\downarrow\downarrow\uparrow\uparrow\right\rangle\right] = \frac{4}{7} \tag{C9}$$
$$P\left[\left|\uparrow\uparrow\uparrow\downarrow\uparrow\downarrow\right\rangle\right] = \frac{1}{28}, P\left[\left|\uparrow\uparrow\downarrow\uparrow\uparrow\downarrow\right\rangle\right] = \frac{1}{28}, P\left[\left|\uparrow\uparrow\uparrow\downarrow\downarrow\uparrow\right\rangle\right] = \frac{1}{28}, P\left[\left|\uparrow\uparrow\downarrow\uparrow\downarrow\uparrow\right\rangle\right] = \frac{1}{28}$$
$$P\left[\left|\uparrow\downarrow\uparrow\uparrow\uparrow\downarrow\right\rangle\right] = \frac{1}{28}, P\left[\left|\downarrow\uparrow\uparrow\uparrow\uparrow\downarrow\right\rangle\right] = \frac{1}{28}, P\left[\left|\uparrow\downarrow\uparrow\uparrow\downarrow\uparrow\right\rangle\right] = \frac{1}{28}, P\left[\left|\downarrow\uparrow\uparrow\uparrow\downarrow\uparrow\right\rangle\right] = \frac{1}{28}$$
$$P\left[\left|\uparrow\downarrow\uparrow\downarrow\uparrow\uparrow\right\rangle\right] = \frac{1}{28}, P\left[\left|\downarrow\uparrow\uparrow\downarrow\uparrow\uparrow\right\rangle\right] = \frac{1}{28}, P\left[\left|\uparrow\downarrow\downarrow\uparrow\uparrow\uparrow\right\rangle\right] = \frac{1}{28}, P\left[\left|\downarrow\uparrow\downarrow\uparrow\uparrow\uparrow\right\rangle\right] = \frac{1}{28}$$

## 2. Periodic boundary conditions

For an AKLT state with PBC, instead of two seperate spins at the boundaries, the MPS has a trace for each matrix product in Eq. (2). The non-trivial basis states for a $L = 2$ AKLT state are:

$$|\psi\rangle_{L=2}^{\text{PBC}} = \alpha_1 \left|+\right\rangle \left|-\right\rangle + \alpha_2 \left|O\right\rangle \left|O\right\rangle + \alpha_3 \left|-\right\rangle \left|+\right\rangle \tag{C10}$$

and

$$\alpha_1 = \text{Tr}\left[A^+ A^-\right] = -\frac{2}{3} \tag{C11}$$
$$\alpha_2 = \text{Tr}\left[A^O A^O\right] = \frac{2}{3}$$
$$\alpha_3 = \text{Tr}\left[A^- A^+\right] = -\frac{2}{3}$$

Again, we substitute $|+\rangle = |\uparrow\uparrow\rangle$, and $|O\rangle = 1/\sqrt{2}\left(|\uparrow\downarrow\rangle + |\downarrow\uparrow\rangle\right)$ into Eq. (C10):

$$|\psi\rangle_{L=2}^{\text{PBC}} = \alpha_1 \left|\uparrow\uparrow\downarrow\downarrow\right\rangle + \frac{\alpha_2}{2}\left(\left|\uparrow\downarrow\uparrow\downarrow\right\rangle + \left|\downarrow\uparrow\uparrow\downarrow\right\rangle + \left|\uparrow\downarrow\downarrow\uparrow\right\rangle + \left|\downarrow\uparrow\downarrow\uparrow\right\rangle\right) + \alpha_3 \left|\downarrow\downarrow\uparrow\uparrow\right\rangle \tag{C12}$$

With the normalization factor $\mathcal{N} = \sqrt{\alpha_1^2 + 4(\alpha_2/2)^2 + \alpha_3^2} = 2/\sqrt{3}$ corresponding to the post-selection process, we obtain the normalized coefficient for the wavefunction as

$$\tilde{\alpha}_1 = \frac{\alpha_1}{\mathcal{N}} = \frac{1}{\sqrt{3}} \tag{C13}$$
$$\tilde{\alpha}_2 = \frac{\alpha_2}{2\mathcal{N}} = \frac{1}{2\sqrt{3}}$$
$$\tilde{\alpha}_3 = \frac{\alpha_3}{\mathcal{N}} = \frac{1}{\sqrt{3}}$$

and the state itself becomes

$$\left|\tilde{\psi}\right\rangle_{L=2}^{\text{PBC}} = \tilde{\alpha}_1 \left|\uparrow\uparrow\downarrow\downarrow\right\rangle + \tilde{\alpha}_2 \left(\left|\uparrow\downarrow\uparrow\downarrow\right\rangle + \left|\downarrow\uparrow\uparrow\downarrow\right\rangle + \left|\uparrow\downarrow\downarrow\uparrow\right\rangle + \left|\downarrow\uparrow\downarrow\uparrow\right\rangle\right) + \tilde{\alpha}_3 \left|\downarrow\downarrow\uparrow\uparrow\right\rangle \tag{C14}$$

Therefore, the normalized probability amplitude for each component basis is then

$$P\left[\left|\uparrow\uparrow\downarrow\downarrow\right\rangle\right] = P\left[\left|\downarrow\downarrow\uparrow\uparrow\right\rangle\right] = \frac{1}{3} \tag{C15}$$
$$P\left[\left|\uparrow\downarrow\uparrow\downarrow\right\rangle\right] = P\left[\left|\downarrow\uparrow\uparrow\downarrow\right\rangle\right] = P\left[\left|\uparrow\downarrow\downarrow\uparrow\right\rangle\right] = P\left[\left|\downarrow\uparrow\downarrow\uparrow\right\rangle\right] = \frac{1}{12}$$

Similarily, for $L = 3$, the state computed from the exact MPS representation is

$$|\psi\rangle_{L=3}^{\text{PBC}} = -\alpha\,|+\rangle\,|O\rangle\,|-\rangle + \alpha\,|+\rangle\,|-\rangle\,|O\rangle + \alpha\,|O\rangle\,|+\rangle\,|-\rangle - \alpha\,|O\rangle\,|-\rangle\,|+\rangle - \alpha\,|-\rangle\,|+\rangle\,|O\rangle + \alpha\,|-\rangle\,|O\rangle\,|+\rangle \qquad \text{(C16)}$$

where $\alpha = \frac{2}{3\sqrt{3}}$. The normalized state is

$$\left|\tilde{\psi}\right\rangle_{L=3}^{\text{PBC}} = -\tilde{\alpha}\left(|\uparrow\uparrow\uparrow\downarrow\downarrow\downarrow\rangle + |\uparrow\uparrow\downarrow\uparrow\downarrow\downarrow\rangle\right) + \tilde{\alpha}\left(|\uparrow\uparrow\downarrow\downarrow\uparrow\downarrow\rangle + |\uparrow\uparrow\downarrow\downarrow\downarrow\uparrow\rangle\right) + \tilde{\alpha}\left(|\uparrow\downarrow\uparrow\uparrow\downarrow\downarrow\rangle + |\downarrow\uparrow\uparrow\uparrow\downarrow\uparrow\rangle\right) \qquad \text{(C17)}$$
$$- \tilde{\alpha}\left(|\uparrow\downarrow\downarrow\downarrow\uparrow\uparrow\rangle + |\downarrow\uparrow\downarrow\downarrow\uparrow\uparrow\rangle\right) - \tilde{\alpha}\left(|\downarrow\downarrow\uparrow\uparrow\uparrow\downarrow\rangle + |\downarrow\downarrow\uparrow\uparrow\downarrow\uparrow\rangle\right) + \tilde{\alpha}\left(|\downarrow\downarrow\uparrow\downarrow\uparrow\uparrow\rangle + |\downarrow\downarrow\downarrow\uparrow\uparrow\uparrow\rangle\right)$$

And then we get the probability amplitude for each basis state in the spin-1/2 basis as

$$P\left[|\downarrow\downarrow\uparrow\uparrow\downarrow\uparrow\rangle\right] = P\left[|\uparrow\downarrow\downarrow\downarrow\uparrow\uparrow\rangle\right] = P\left[|\downarrow\uparrow\uparrow\uparrow\downarrow\downarrow\rangle\right] = P\left[|\uparrow\uparrow\downarrow\uparrow\downarrow\downarrow\rangle\right] = P\left[|\uparrow\uparrow\downarrow\downarrow\downarrow\uparrow\rangle\right] = P\left[|\downarrow\downarrow\uparrow\uparrow\uparrow\downarrow\rangle\right] \qquad \text{(C18)}$$
$$= P\left[|\downarrow\downarrow\uparrow\downarrow\uparrow\uparrow\rangle\right] = P\left[|\downarrow\downarrow\downarrow\uparrow\uparrow\uparrow\rangle\right] = P\left[|\downarrow\uparrow\downarrow\downarrow\uparrow\uparrow\rangle\right] = P\left[|\uparrow\downarrow\uparrow\uparrow\downarrow\downarrow\rangle\right] = P\left[|\uparrow\uparrow\uparrow\downarrow\downarrow\downarrow\rangle\right] = P\left[|\uparrow\uparrow\downarrow\downarrow\uparrow\downarrow\rangle\right] = \frac{1}{12}$$

We show the results for the $L = 2$ and $L = 3$ AKLT states with PBC in FIG. 10. In order to host a minimal number of single-qubit as well as $CX$ gates, the implementation for PBC requires a special circuit geometry where the both edge spins are connected, as shown in FIG. 10(a). We plot the results for PBC AKLT state probability distribution for $L = 2$ [FIG. 10(b)] and $L = 3$ [FIG. 10(c)] using the noiseless `aer_simulator` from Qiskit, which shows good agreement with the exact results calculated from MPS, as derived above in Sec . C 2.

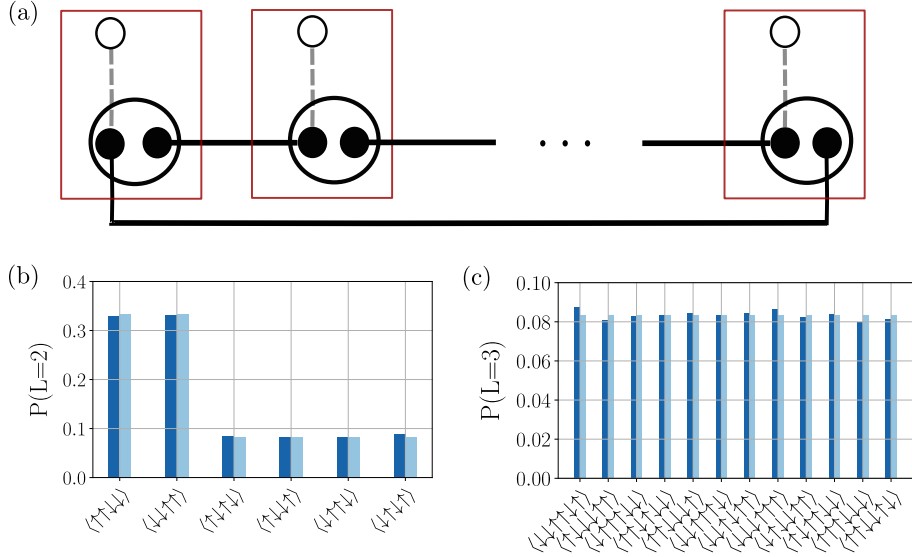

FIG. 10. AKLT state preparation with periodic boundary condition (PBC): (a) Setup for AKLT state with PBC. Each solid black dot indicates a physical qubit site, and the solid black line represents the initial singlet bond. Two ancilla qubits are represented by smaller hallow circles, connected to the corresponding physical qubit via a dashed grey line. The larger circles represent the spin-1 site. The unitary operators are applied on the three-qubit sites (red-lined squares); (b) Probability amplitude for $L = 2$ and 3 AKLT state with PBC. Lighter blue bar represents the exact values calculated from MPS representation in Sec . C 2, and the darker blue bar represents the results obtained from the noiseless `aer_simulator` in Qiskit.

## Appendix D: Readout error mitigation and device specifications

A major error which can be mitigated in our experiment on IBM Q is the readout error, where there exists a possibility of measuring $|\uparrow\rangle$ but renders a $|\downarrow\rangle$, and vice versa. Recent progress have seen tremendous efforts in the mitigation of the measurement error [84, 128–133]. For Qiskit [63] environment itself, one could first run a number of calibration circuits with different initial conditions, and then estimate the true measurement counts based on the calibration matrix formed from the outcomes from those calibration circuits [134, 135]. In this paper, we utilize a

recent readout error mitigation approach [91] which requires only a handful of circuits without the construction of full calibration matrix.

In order to be suitable for the job submission framework of IBM Q platform, and to make full use of the calibration approach, we combine the circuits ('physical circuits') for the preparation of AKLT state with the calibration circuits together into one single job and submit to the IBM Q platform on the cloud. This is to enforce that the 'physical circuits' and the calibration circuits are executed almost at the same time, which will make the calibration more accurate. Also, in order to have the same quantum register layout for both 'physical circuits' and the calibration circuit, we first select and transpile the 'physical circuit' onto the corresponding real device with respect to the best fitness function using the device error data which were calibrated by IBM Q for high-difelity quantum nondemolition (QND) measurements [17], and then use this particular layout for the calibration circuit so that the qubits used for both categories of circuits are exactly the same. We then submit both categories of circuits together to the real device on IBM Q for execution.

We show the device error obtained from IBM Q *ibmq_montreal* in FIG. 11.

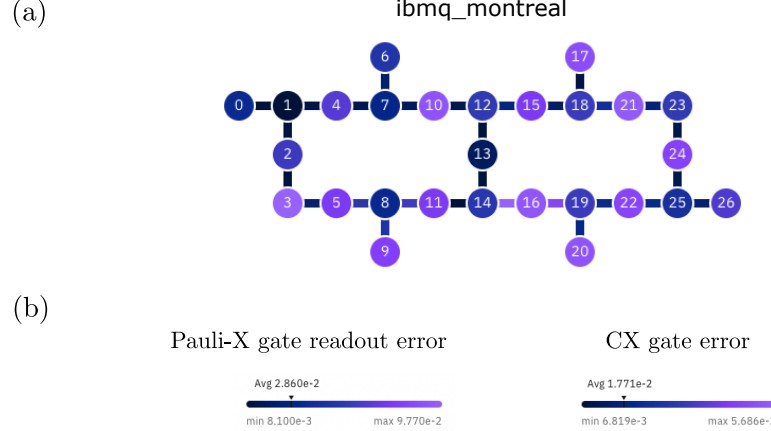

FIG. 11. Calibration data of *ibmq_montreal* on 2022-09-22 22:56: (a) Mapview of the calibration data on IBM Q *ibmq_montreal* device; (b) Range of single-qubit Pauli-$X$ gate error (left panel) and $CX$ gate error (right panel) with their averaged values. The averaged relaxation time $T_1$ and decoherence time $T_2$ for the qubits is $122.93\mu s$ and $92.16\mu s$, respectively.