# Peer review of "High-fidelity realization of the AKLT state on a NISQ-era quantum processor"

_SciPost Physics_

## Round 2 · Referee Report · Emanuele Dalla Torre (Referee 1) · 2023-3-29

Strengths

1- The paper addresses an interesting problem, i.e. how to create the ground state of a one-dimensional many-body quantum Hamiltonian, known as the AKLT state. 2- The method is non-conventional, as it uses ancilla qubits to effectively induce a non-unitary operation on the system. 2- The method relies on and implements an efficient variational algorithm. 3- The authors demonstrate their approach in both noiseless simulations and real calculations on superconducting quantum computers. 4- The article is well motivated, concise, and clearly written.

Weaknesses

Weaknesses 1- The approach is not compared with other state-of-the art state preparation algorithms 2- The fidelity measured in the real hardware is inaccurate

Report

This article describes a method to create the AKLT state, a notable one-dimensional quantum state, on a quantum computer. Following to the original guidelines described by AKLT, the state is created by first creating singlet states between alternating pairs of neighboring qubits ($2n,2n+1$ for integer $n$), and then projecting this state on the spin-1 subspace of the other pairs ($2n-1,2n$), see Fig. 1a of the paper. The AKLT state is known to be the ground state of a specific one-dimensional Hamiltonian, Eq. (5) of the paper, and is the simplest example of valence bond solid (VBS). The key difficulty of this receipt is who to realize the projection operation, which is a non-unitary process. The authors propose and demonstrate how to perform this step using a single ancilla qubit for each pair of qubits. They implement the algorithm on an IBM quantum computer and claim to achieve very high fidelities. As detailed in the “weaknesses” and “requested changes” parts, the article deserves significant revision.

Requested changes

1- As the author mention in the introduction, several earlier works developed direct methods to implement matrix-product states (MPS) using unitary gates, see Refs. [27-29], without the need to insert additional ancillas. The number of gates requires scales approximately as $L 4^\chi$, where $L$ is the number of qubits and $\chi$ the entanglement entropy of the state. The AKLT state (after projection to the spin-1 subspace) can be written as an MPS with $\chi=log2(2)$, and hence I would expect $4L$ gates to be sufficient to exactly create this state. Hence, I expect this approach to require less gates than the one by the present authors ($9(L/2)$), less qubits ($2L$ instead of $3L$), and avoid the hassle of post-selection. While the present approach may still be interesting because it allows an easy extension to more complex VBS states, I think that such comparison should be carried out explicitly. To summarize, I request that the author use the methods of Ref. [27-29] to create the AKLT state directly without ancillas and compare the efficiency of this appraoch with theirs.

2- The authors use two distinct measures to probe the fidelity of the AKLT state, in theory and in practice. For the theoretical analysis, they consider the fidelity defined in Eq. (16), while for the practical implementation the use Eq. (18) (Hellinger fidelity). I have some problems with both definitions. 2A- Concerning the first definition, Eq. (16), it is not clear to me why they use random states, rather than a complete set of states, i.e. perform a full state tomography. In the case of 3 qubits, Eq. (16) requires 64 measurements and is easily realizable. I request that the authors compare the two methods, namely Eq. (64) with random states and full state tomography. Presumably, the two approaches should give the same result. Also, I strongly suggest that they show the results of the fidelity as a function of layers, Fig. 9, in the main text, as this is one of their main results. Finally, why do 9 layers give a worse result than 8? The authors claim that this is “due to the other parameters chosen”, but I could not understand what they mean here. 2B- Concerning the second definition, Eq. (18), while this approach may be useful to describe other states, it is not adequate to the AKLT state. The reason is that, as shown in Eq. (C9), the AKLT state has equal probabilities to be found in all “legal” states. The entanglement properties of the AKLT state are stored in the complex phases of the wavefunction, which are not probed by Eq. (18). As a consequence, when the mitigation protocol is applied to a maximally mixed state (an equal probability of all outcomes), Eq. (18) would deliver perfect fidelity, in spite of the exponentially small overlap between the state and the target one. In my opinion, this is the reason of the very large fidelity observed in Fig. 6 for large $L$. Indeed, for quantum gates with 1% error (after mitigation) and a circuit with $L=5$ qubits with $N=5*9=45$ gates, one would expect a fidelity that is smaller than 50%, even after mitigation. To address this point, I request that the authors consider additional measures of fidelity that are applicable to mid-size circuits, such as shadow tomography or cross-entropy benchmarking. Importantly, they should make sure that the definition of fidelity that they use does not return a large value for a maximally mixed state.

Minor comments 3- The authors write in the introduction that “state preparation is a fundamentally non-unitary process which requires the implementation of non-unitary operators.” . This sentence seems to me incorrect or misleading as any pure quantum state can be generated using unitary operations starting from the |0000> state. Am I missing something? 4- Page 2: “prepar” --> “prepare” 5- Page 5: “a few hundred” --> “a few tens”. Superconducting quantum circuits have typical volume of 32-64 and do not allow to apply so many gates. Most demonstrations using this system involve up to a few tens of gates at most.

  • validity: good
  • significance: good
  • originality: high
  • clarity: high
  • formatting: excellent
  • grammar: excellent

Author:  Tianqi Chen  on 2023-07-31  [id 3857]

(in reply to Report 1 by Emanuele Dalla Torre on 2023-03-29)
Category:
answer to question

We thank Prof. Emanuele Dalla Torre for his careful reading of our manuscript. Our responses to each of the raised points are as below.

The referee writes: “This article describes a method to create the AKLT state, a notable one-dimensional quantum state, on a quantum computer. Following to the original guidelines described by AKLT, the state is created by first creating singlet states between alternating pairs of neighboring qubits (2n, 2n+1 for integer n), and then projecting this state on the spin-1 subspace of the other pairs (2n−1, 2n), see Fig. 1a of the paper. The AKLT state is known to be the ground state of a specific one-dimensional Hamiltonian, Eq. (5) of the paper, and is the simplest example of valence bond solid (VBS). The key difficulty of this receipt is who to realize the projection operation, which is a non-unitary process. The authors propose and demonstrate how to perform this step using a single ancilla qubit for each pair of qubits. They implement the algorithm on an IBM quantum computer and claim to achieve very high fidelities. As detailed in the “weaknesses” and “requested changes” parts, the article deserves significant revision.”

Our responses: We would like to thank Prof. Emanuele Dalla Torre for his careful reading, and we agree that we should give further details to more strongly support our claim of very high fidelities.

The referee writes: “1-As the author mention in the introduction, several earlier works developed direct methods to implement matrix-product states (MPS) using unitary gates, see Refs. [27-29], without the need to insert additional ancillas. The number of gates requires scales approximately as L 4 χ , where L is the number of qubits and χ the entanglement entropy of the state. The AKLT state (after projection to the spin-1 subspace) can be written as an MPS with χ = l o g 2 ( 2 ) , and hence I would expect 4 L gates to be sufficient to exactly create this state. Hence, I expect this approach to require less gates than the one by the present authors ( 9 ( L / 2 ) ), less qubits ( 2 L instead of 3 L ), and avoid the hassle of post-selection. While the present approach may still be interesting because it allows an easy extension to more complex VBS states, I think that such comparison should be carried out explicitly. To summarize, I request that the author use the methods of Ref. [27-29] to create the AKLT state directly without ancillas and compare the efficiency of this appraoch with theirs.”

Our responses: We thank Prof. Emanuele Dalla Torre for raising this point regarding using existing MPS-based state preparation methods. Thus, we carefully check this point: whether the AKLT state (either in spin-1 or spin-1/2 representation) with open boundary condition could be directly prepared using the existing methods in Ref. [S.-J. Ran (2020); A. Holmes and A. Y. Matsuura (2020); S.-H. Lin et al. (2021)] during the initial stage of this project. Although these existing MPS/tensor network-based methods do not require the post-selection, they could not be directly applied to the preparation of the AKLT state with our setup in spin-1/2 representation. The reasons are two-fold: first, for Ref [A. Holmes and A. Y. Matsuura (2020); S.-H. Lin et al. (2021)], the operator corresponding to each physical dimension (d=2) is required to be unitary so that it can be transformed directly into single- or two-body qubit gates, while for us, in the spin-1/2 representation, the projection operator itself is non-unitary and therefore it is not possible to do so. Second, even if we were to work in the spin-1 representation, in Ref. [S.-J. Ran (2020)], although they indeed gave a relatively general description of the MPS-based approach where d can be larger than 2, they require the auxiliary bond dimension (D) to be at least equal or larger than the local physical dimension, i.e. D>=d. As the referee mentioned, after projection to the spin-1 subspace, the AKLT state has a local physical dimension d=3 with auxiliary bond dimensions equal to 2 (see Eq. (4) of the manuscript). This means that the method proposed in Ref. [S.-J. Ran (2020)] may not be applied directly to our setup for the AKLT state preparation.

More importantly, we would also like to point out that the scope of this work is not on developing new MPS-based methods to prepare the AKLT state, but rather, inspired by the MPS representation of the AKLT state, we introduce and discuss an algorithm to implement and characterize the AKLT state on a NISQ-era quantum device, which has only been covered very little among other similar works most recently [K. C. Smith et al. (2023); B. Murta, P. M. Q. Cruz and J. Fernandez-Rossier (2023)]. Also, if we were to use the tensor network-based methods such as in Ref. [S.-J. Ran (2020); A. Holmes and A. Y. Matsuura (2020); S.-H. Lin et al. (2021)] without any post-selection, as the target state is to be prepared on a qubit-based quantum processor, the spin-1 representation would need to be explicitly expanded to the spin-1/2 representation first (as in Appendix C1). From there, by a direct summation of all the local tensors on each physical site, it might be possible to construct an MPS formalism where the auxiliary bond dimension is larger than the local dimension, which may allow us to follow the recipe introduced in Ref. [S.-J. Ran (2020)]. We currently plan to work on this in our future work for the improvement of the AKLT state preparation. To make the above points clearer, we therefore have expanded the discussion and have added a number of lines (highlighted in blue) in both the introduction and the conclusion part to address this accordingly.

The referee writes: “2- The authors use two distinct measures to probe the fidelity of the AKLT state, in theory and in practice. For the theoretical analysis, they consider the fidelity defined in Eq. (16), while for the practical implementation the use Eq. (18) (Hellinger fidelity). I have some problems with both definitions. “

Our responses: We thank Prof. Emanuele Dalla Torre for pointing out his problem with the definitions, because it made us realize that some parts of the manuscript may not have been presented clearly enough. First, we would like to point out that Eq. (16) and Eq. (18) are two different definitions of the fidelity, not “two distinct measures to probe the fidelity of the AKLT state”. In fact, we only consider one fidelity to evaluate how well the AKLT state is prepared on IBM Q in our manuscript, which is Eq. (18).

The calligraphic F from Eq. (16) is for the variational algorithm of the circuit recompilation (Sec. 3.3), which corresponds to the fidelity between the original target operator U, and the recompiled operator V, both being acted upon a random state |β⟩. This is a general approach to identifying trained circuits in the variational algorithm (see PRX Quantum 2, 010317 (2021) and arXiv:1807.0080). The quantity F in Eq. (16) corresponds to how close the recompiled operator (V) is to the original unitary operator (U) using the MPS-based variational algorithm on a classical computer. Then, the normal F from Eq. (18) in Sec. 4.1 is the actual (Hellinger) fidelity of the prepared AKLT state which is obtained from executing the recompiled operator (V) on a real IBM Q device. To clarify all these points, we have thus called Eq. (16) the circuit fidelity, and Eq. (18) Hellinger fidelity throughout the whole manuscript.

The referee writes: “2A- Concerning the first definition, Eq. (16), it is not clear to me why they use random states, rather than a complete set of states, i.e. perform a full state tomography. In the case of 3 qubits, Eq. (16) requires 64 measurements and is easily realizable. I request that the authors compare the two methods, namely Eq. (64) with random states and full state tomography. Presumably, the two approaches should give the same result. Also, I strongly suggest that they show the results of the fidelity as a function of layers, Fig. 9, in the main text, as this is one of their main results. Finally, why do 9 layers give a worse result than 8? The authors claim that this is “due to the other parameters chosen”, but I could not understand what they mean here. ”

Our responses: We apologize that we did not make it clearer in our previous manuscript, which might cause some confusion. As explained above, since Eq. (16) and Eq. (18) refer to distinct concepts, it would be difficult to compare them. Nevertheless, we sincerely appreciate the valuable feedback from the referee. As the primary focus of this work revolves around the characterization of the AKLT state on a NISQ-era quantum processor, we have carefully considered the referee's suggestion regarding the inclusion of Fig. (9) in the main text. Upon reflection, we concur with the referee that this figure primarily pertains to the performance analysis of the variational algorithm for circuit recompilation, and as such, may not directly align with our major results. Therefore, we will gracefully omit Fig. (9) from the main text and keep it in the appendix to enhance the coherence and clarity of our core findings. Furthermore, we would like to emphasize that all the random states utilized in our study comprehensively cover full states in the computational basis. As a result, the effect of our method for identifying trained circuits is indeed having the same effect compared with the one with full-state tomography. This point can be verified by the excellent agreements between the exact and noiseless results shown in FIG. 10, where the slight difference is the result of sampling noise in the Aer simulator from the Qiskit. We acknowledge the referee's valuable input, and we ensure that this point is appropriately articulated in the revised manuscript.

Finally, as shown in Fig. 9 (b), both 8-layer and 9-layer results have shown that they can lead to the fidelity being equal to one. However, from panel (a), it just means that for 9 layers, other parameters being equal, after the optimizations, on average the fidelity between the target circuit and the recompiled ansatz circuit is not as good as the one obtained with 8 layers. This is because the 9-layer ansatz circuit has more parameters than the one with 8 layers, and hence one would have to tune other parameters in the optimization to achieve the same performance, such as increasing the number of basin hopping so that for 9 layers, it will not be stuck in the local minimal. In the revised manuscript, we have specified what other parameters to possibly tune for the optimization of the circuit in this paragraph.

The referee writes: “2B- Concerning the second definition, Eq. (18), while this approach may be useful to describe other states, it is not adequate to the AKLT state. The reason is that, as shown in Eq. (C9), the AKLT state has equal probabilities to be found in all “legal” states. The entanglement properties of the AKLT state are stored in the complex phases of the wavefunction, which are not probed by Eq. (18). As a consequence, when the mitigation protocol is applied to a maximally mixed state (an equal probability of all outcomes), Eq. (18) would deliver perfect fidelity, in spite of the exponentially small overlap between the state and the target one. In my opinion, this is the reason of the very large fidelity observed in Fig. 6 for large L . Indeed, for quantum gates with 1% error (after mitigation) and a circuit with L = 5 qubits with N = 5 ∗ 9 = 45 gates, one would expect a fidelity that is smaller than 50%, even after mitigation. To address this point, I request that the authors consider additional measures of fidelity that are applicable to mid-size circuits, such as shadow tomography or cross-entropy benchmarking. Importantly, they should make sure that the definition of fidelity that they use does not return a large value for a maximally mixed state.”

Our responses: We thank Prof. Emanuele Dalla Torre for this important perspective because it made us realize that the Hellinger fidelity used in this manuscript, although being intuitive and suitable for the IBM Q device analogue to the classical probability distribution, may not precisely capture the entanglement properties from the prepared state on IBM Q, as it does take the coherent information of the basis states into account. To this end, we have added a completely new section in the appendix (Appendix E, highlighted in blue) on themeasurement of the entanglement spectrum using the expansion of the density matrix in terms of Pauli strings for L=2, and we have also expanded the discussion in Sec. 4.1 and 4.2 by adding several lines (highlighted in blue). We shall carry out our explanation on how to measure the entanglement properties with the current setup in our manuscript sequentially with the flow of our reply below.

First, we agree with the referee that using the Hellinger fidelity alone may not capture the off-diagonal components from the prepared state density matrix. Hence the fidelity shown in Fig. 6 for large L after the error mitigation is large, as the overlap between the off-diagonal components are not considered. But the reason why we adopted this fidelity measurement is that this quantity is suitable for IBM Q and it is straightforward to calculate and compared it with the noiseless value. Also, on the plus side, this circumvents the high number of tomography circuits. This is because one could only obtain the number of shots for each outcome of the measurements on IBM Q, and they essentially constitute the diagonal components of the prepared state density matrix.

Now, allow us to point out that among all the other two works which proposed to implement the AKLT state on IBM Q [Kevin et al. (2023); Murta et al. (2023)], only Ref. [Kevin et al. (2023)] can perform the complete tomography of the prepared AKLT state on IBM Q, and obtain the full density matrix of the AKLT state in spin-1 representation (the reason being elaborated below).

Thus, we carefully think about whether the shadow tomography or cross-entropy benchmarking is suitable for our model, and we explain why we cannot implement the above in our setup, as the following two-fold reasons:

First, the tomography approaches would require extra circuits after the preparation circuit, and in our case, this would introduce additional error (gate fidelity) which could hinder the successful preparation of the AKLT state, since we are using two spin-1/2 plus an additional ancilla qubit for the preparation of the state on each spin-1 site, and the system itself is already quite large.

Second, compared with the encoding of the triplet states in Ref. [Kevin et al. (2023)], our encodings are the same as the original definition of the AKLT state: , while Ref. [Kevin et al. (2023)] introduced a different set of encodings: |+⟩=|10⟩,|-〉=|01⟩,|O⟩=|00⟩. Also, since their approach does not require any ancilla qubit and post-selection, this allows them to prepare small fragments of the total chain and combine them using Bell measurements, and eventually, they could perform the quantum tomography of the state and obtain the density matrix in the spin-1 basis.

In this case, we consider a more straightforward way in the spin-1/2 representation: using the Pauli string expansion of the density matrix from the measurements on IBM Q itself.

As of July 2023, the device which we used to produce all the results in our manuscript (ibmq_montreal) has already been retired, and instead, we could only perform our tomography experiment on another device (we have chosen ibm_algiers due to good qubit gates fidelity and small readout error). We have therefore added a section on the entanglement spectrum for L=2 in Appendix E, and we have also expanded the discussion in Sec. 4.1 and 4.2. We compare and check its fidelity with both the noiseless and the exact results of the prepared AKLT state in our manuscript. Our new results show that indeed due to the off-diagonal components in the reduced density matrix, there is a conceivable difference between the exact results and those from the real device.

The referee writes: “Minor comments 3- The authors write in the introduction that “state preparation is a fundamentally non-unitary process which requires the implementation of non-unitary operators.” . This sentence seems to me incorrect or misleading as any pure quantum state can be generated using unitary operations starting from the |0000> state. Am I missing something?”

Our responses: We thank Prof. Emanuele Dalla Torre for pointing this out to us. Indeed this sentence is not accurate. Perhaps putting it in the context of our research here would help. What we intended to express here is that there exists a variety of states which need non-unitary operators to get prepared, e.g., for the realization of the fractional quantum Hall (FQH) state with toroidal boundary conditions [Nakamura, Wang & Bergholtz (2012); Rahmani et al. (2020)], it is essentially constructed by a non-unitary projection operator acting on the root configuration to obtain the resulting FQH state. Also, in the non-equilibrium setup, an external source is needed to prepare the state within the system, such as a random subnormalized two-level state [A. W. Schlimgen et al. (2022)], and this process requires a non-unitary operator to be implemented on a quantum device.

To this end, we have changed this sentence to “…state preparation with projective operations to generate it…” which is highlighted in blue, and added the above additional references as well.

The referee writes: “4- Page 2: “prepar” --> “prepare” ”

Our responses: We thank the referee for pointing this out. We have corrected the spelling.

The referee writes: “5- Page 5: “a few hundred” --> “a few tens”. Superconducting quantum circuits have typical volume of 32-64 and do not allow to apply so many gates. Most demonstrations using this system involve up to a few tens of gates at most. ”

Our responses: We thank the referee for the clarification. Indeed for a superconducting system such as IBM Q, there is a limit for the number of gates applied to it. We have corrected this in our manuscript.

---

## Round 2 · Referee Report · Anonymous (Referee 2) · 2023-6-21

Strengths

1) The paper is well written and well presented 2) The authors proposed a new algorithm to prepare a AKLT state 3) The algorithm is very versatile and it can be applied for different VBS states

Weaknesses

1) The approach is not compared with other algorithms

Report

The authors present an efficient algorithm for preparing the AKLT state on an IBM quantum computer, focusing on NISQ-era quantum processors. The algorithm utilizes an additional ancilla qubit to embed a non-unitary projection operator into a unitary operator acting on three qubits. By employing variational recompilation, the three-qubit operation is transformed into a parametrized circuit with reduced circuit depth.

The algorithm involves evolving a trivial initial product state of singlets along with ancilla qubits in a spin-up state using the parametrized circuit. The AKLT state is obtained through post-selection and spin number conservation.

The approach presented in the paper offers several advantages. It enables efficient preparation of the AKLT state on NISQ-era quantum processors by reducing the number of CX and single-qubit gates required. The variational recompilation optimizes the quantum circuit, mitigating the impact of gate errors.

Furthermore, this algorithm can be used for higher-dimension AKLT state and for other types of VBS states that are very important in the context of condensed matter.

I would recommend the publication of this manuscript in Scipost. However, I have some questions for the authors that I would like them to address before proceeding with the publication.

Requested changes

1) In the last years many algorithms and approaches based on tensor networks have been proposed for state preparation. Could you compare the efficiency of the proposed algorithm with the efficiency of a TN based algorithm?

2) To further validate the prepared state, could you measure its entanglement spectrum and compare it against known theoretical results for the AKLT state?

3)The error mitigation method used by the authors is focused on the readout error. In this case, the improvement in the fidelity value is modest. There are other error mitigation techniques like zero-noise extrapolation or probabilistic error cancellation. Is it possible to use another technique to improve the results?

  • validity: high
  • significance: high
  • originality: good
  • clarity: high
  • formatting: excellent
  • grammar: good

Author:  Tianqi Chen  on 2023-07-31  [id 3858]

(in reply to Report 2 on 2023-06-21)
Category:
answer to question

We thank the referee for his/her careful reading of our manuscript. Our responses to each of the raised points are as below.

The referee writes:
“The authors present an efficient algorithm for preparing the AKLT state on an IBM quantum computer, focusing on NISQ-era quantum processors. The algorithm utilizes an additional ancilla qubit to embed a non-unitary projection operator into a unitary operator acting on three qubits. By employing variational recompilation, the three-qubit operation is transformed into a parametrized circuit with reduced circuit depth.
The algorithm involves evolving a trivial initial product state of singlets along with ancilla qubits in a spin-up state using the parametrized circuit. The AKLT state is obtained through post-selection and spin number conservation.
The approach presented in the paper offers several advantages. It enables efficient preparation of the AKLT state on NISQ-era quantum processors by reducing the number of CX and single-qubit gates required. The variational recompilation optimizes the quantum circuit, mitigating the impact of gate errors.
Furthermore, this algorithm can be used for higher-dimension AKLT state and for other types of VBS states that are very important in the context of condensed matter.
I would recommend the publication of this manuscript in Scipost. However, I have some questions for the authors that I would like them to address before proceeding with the publication.”

Our responses:
We thank the referee for his/her summary as well as the positive evaluation of the manuscript by writing ‘The approach presented in the paper offers several advantages’, ‘enables efficient preparation of the AKLT state on NISQ-era quantum processors’, and we thank him/her for ‘recommend the publication of this manuscript in Scipost’.

The referee writes:
“1) In the last years many algorithms and approaches based on tensor networks have been proposed for state preparation. Could you compare the efficiency of the proposed algorithm with the efficiency of a TN based algorithm?”

Our responses:
We thank the referee for his/her comment. This point has also been mentioned in another referee report. A general remark on this question is that compared with our approach, most tensor network (TN)-based methods do not require additional ancilla qubits (additional physical sites) and post-selections upon the measurements (contraction of the tensors to compute the observables), and therefore their way of the state preparation is deterministic, and the number of gates required is linear to the system size L (in spin-1 representation). Whereas for our approach, the number of gates is proportional to 3L (in spin-1/2 representation). But as we have written in the manuscript, the AKLT state could be written in an exact representation of MPS, or one-dimensional tensor train form, as illustrated in Eq. (2) (PBC), and Eq. (3) (OBC). For each tensor A^(σ_i ) on each physical site i, the physical dimension d=3, and the auxiliary bond dimension M=2. Therefore, it is not capable of directly applying the TN-based methods in Ref. [27-29] to prepare the AKLT state, as it requires M≥d. This triggered us to think of another way to prepare the AKLT state by projecting the singlet product states into the spin-1 manifold, which is the focus of our work, and it involves the implementation of a non-unitary operator. A recent work in Ref. [57] has prepared the 1D AKLT state adiabatically with tensor networks, but it is not clear how to directly implement it on IBM Q or other NISQ-era platforms.
If we were to use the approaches introduced in Ref. [27-29], the spin-1 representation would need to be explicitly expanded to the spin-1/2 representation first (as in Appendix C1), and from there by a direct summation of all the local tensors on each physical site, it might be possible to construct an MPS formalism where the auxiliary bond dimension is larger than the local dimension, which may allow us to follow the recipe introduced in Ref. [27]. We plan to work on this in our future work for the improvement of the AKLT state preparation. We have updated our manuscript by adding several lines on a brief clarification of why other MPS-based state preparation approach of the AKLT state could not be directly applied in our setup in the introduction in the introduction part in our revised manuscript.

The referee writes:
“2) To further validate the prepared state, could you measure its entanglement spectrum and compare it against known theoretical results for the AKLT state?”

Our responses:
We thank the referee for his/her perspective. We have done additional calculations, and our simulations of the entanglement spectrum (ES) are shown in Appendix E. The ES provide a good characterization of the AKLT state. To measure the ES, one first needs to extract a subsystem, and obtain the reduced density matrix of this subsystem. The ES can then be determined from the eigenvalues of this matrix. When it comes to simulations on quantum computers, directly measuring the elements in a reduced-density matrix proves challenging. A potential method is discussed in Phys. Rev. Lett. 121, 086808, where the symmetry-protected topological (SPT) state can be mapped to our AKLT states (as referenced in PRB 96, 165124). Following this method, we measure the reduced density matrix of a two-qubit subspace for a system with an ancilla and four physical qubits. Here these two qubits are built by the first two physical qubits. To carry out our simulations, we first expand this reduced density matrix in terms of the Pauli string, as described in Eq. 39, and then after obtaining the coefficients in Eq. 40, we add up all the Pauli strings. The simulation results are shown in FIG. 12. For the density plots in FIG. 12(a), we find that all results exhibit good agreements, and even if there is no readout error mitigation, the error induced by the device noise is not significant. For the ES in FIG. 12(a), there exhibits a slight difference between the noisy and noiseless results. The reason is that for our 2-qubit subspace, we need to add up 16 noisy outcomes (16 Pauli strings), which leads to slight errors in the reduced density matrix. However, the eigenvalues of this reduced density matrix are extremely sensitive to elements, and even minor errors can lead to poor measurement.

The referee writes:
“3)The error mitigation method used by the authors is focused on the readout error. In this case, the improvement in the fidelity value is modest. There are other error mitigation techniques like zero-noise extrapolation or probabilistic error cancellation. Is it possible to use another technique to improve the results?”

Our responses:
We thank the referee for his/her remark on the error mitigation approach used in our manuscript. In our work, except for the error induced by the CX gate noise, a critical error is from the readout error, and especially, the readout error on the ancilla qubit affects the results significantly. In short words, zero-noise extrapolation (ZNE) requires that one can execute the same circuit under different noise levels, and probabilistic error cancellation (PEC) requires that one can execute a series of linear circuits at the same level of hardware noise. But both require that one has got full tomographic knowledge of the hardware gates. Indeed, these two methods are powerful in error mitigation, but we find that these two methods are not suitable for our circuits. For the ZNE, with the IBM Q device, live noise data for each simulation isn't accessible; we can only retrieve noise data measured by the IBM team a few hours before our experiments. This makes constructing an accurate noisy model challenging. As for PEC, aside from the issue of accessing noise data, we also cannot ensure consistent noise levels across all simulations. Furthermore, our Variational Quantum Algorithm (VQA) circuits have a fixed size; we can't adjust the circuit depth as in the traditional Trotter step approach. Additionally, full tomographic knowledge of the hardware gates is unavailable to private users on the IBM Q device. Thus, implementing ZNE and PEC for our simulations becomes highly difficult.

---

## Round 2 · Referee Report · Anonymous (Referee 3) · 2023-7-2

Report

The authors implement and study the fidelity of the Affleck-Kennedy-Lieb-Tasaki (AKLT) state on ibm_montreal, a noisy intermediate-scale quantum (NISQ) processor. To implement the necessary projections, they augment each spin-1 site by an ancilla qubit such that each spin 1 is represented by 3 spins 1/2, and the projection on the spin-1 subspace is realized by post-selection on the ancilla qubit. The projection involves an $\hat{\cal U}$ operator that would require a high-depth circuit to be implemented on the device. The authors thus approximate it by a significantly shallower variational $\hat{\cal V}$ circuit.

Overall, this appears to be solid and interesting work. If I have any objection, the effort required on a classical computer (e.g., to optimize $\hat{\cal V}$) seems quite high to implement an AKLT chain even with a modest size of $L \le 5$ spins 1 and open boundary conditions. The authors might thus wish to reinforce their comments on perspectives of their work, e.g., in case devices with more qubits should become available. A related "detail" is that in my opinion the authors should specify their optimized circuit $\hat{\cal V}$ in order to permit others to reproduce their work.

I have several further technical comments on some presentation details that I list under "Requested changes".

Requested changes

Order reflecting appearance, not importance: 1- The "for the first time" in the abstract is a claim to priority that in my opinion is neither necessary nor appropriate. I thus recommend removing it. 2- Attributions in section II are a bit strange. Firstly, the original articles [34,35] are only cited at the very end of this section. Secondly, a matrix-product representation of the AKLT state appears already in the review article H.-J. Mikeska and A.K. Kolezhuk, One-Dimensional Magnetism, Lect. Notes Phys. 645, 1-83 (2004) that predates Ref. [65]. 3- The first line of Eq. (6) could be read either as $|+\rangle (\langle + | + |O\rangle)( \langle O | + |-\rangle) \langle - |$ or as $(|+\rangle \langle + |) +( |O\rangle \langle O |) + (|-\rangle \langle - |)$. I recommend adding brackets in order to avoid any possible confusion. 4- Beginning of section III C: the authors say that the default transpile function from Qiskit yields 24 $CX$ gates to represent the unitary $\hat{\cal U}$. Just before, they say that a circuit with more than a few hundred $CX$ gates would not be ideal. I think that this combination of the two statements is confusing, and recommend clarifying this discussion. 5- In relation to Fig. 3: the authors should make their optimized circuit $\hat{\cal V}$ accessible. 6- Fig. 4: the color scales were too small to be readable in my printout. 7- As far as I can see, Refs. [33,49,53-55,110,117,133] have been published in the meantime. I do realize that at least in some cases, the references were still unpublished when the present work was submitted, but I still recommend that the authors use the opportunity to update their references when they resubmit. 8- Remove the spurious "july 2011 Special Issue" from Ref. [46]. 9- Ref. [70] is a duplicate of Ref. [65]. 10- Ref. [103] lacks a DOI. 11- Remove the "in Sec. C2" above Fig. 10 (this is an unnecessary self-reference of appendix C2 to itself). 12- Fig. 11: the color scales were too small to be readable in my printout. 13- It would be nice if the authors could use the appropriate SciPost Physics style files when they resubmit.

  • validity: high
  • significance: high
  • originality: high
  • clarity: high
  • formatting: excellent
  • grammar: excellent

Author:  Tianqi Chen  on 2023-07-31  [id 3859]

(in reply to Report 3 on 2023-07-02)
Category:
answer to question

We thank the referee for his/her careful reading of our manuscript. Our responses to each of the raised points are as below.

The referee writes: “The authors implement and study the fidelity of the Affleck-Kennedy-Lieb-Tasaki (AKLT) state on ibm_montreal, a noisy intermediate-scale quantum (NISQ) processor. To implement the necessary projections, they augment each spin-1 site by an ancilla qubit such that each spin 1 is represented by 3 spins 1/2, and the projection on the spin-1 subspace is realized by post-selection on the ancilla qubit. The projection involves an U operator that would require a high-depth circuit to be implemented on the device. The authors thus approximate it by a significantly shallower variational V circuit. Overall, this appears to be solid and interesting work. If I have any objection, the effort required on a classical computer (e.g., to optimize V) seems quite high to implement an AKLT chain even with a modest size of L≤5 spins 1 and open boundary conditions. The authors might thus wish to reinforce their comments on perspectives of their work, e.g., in case devices with more qubits should become available. A related "detail" is that in my opinion the authors should specify their optimized circuit V in order to permit others to reproduce their work. I have several further technical comments on some presentation details that I list under "Requested changes".”

Our responses: We are very grateful to the referee for his/her summary of our manuscript, as well as the positive evaluation and comments of it by justifying our work as ‘solid and interesting’. As the referee has mentioned, indeed at this juncture the required efforts to realize the AKLT state for a longer chain seem to be high for the NISQ-era device, and therefore a better fidelity of the preparation of the state shall be expected when more qubits with high gate fidelity are available in the future. We have modified and added this into the corresponding section in our manuscript. As for the optimized circuit V, we have made the dataset for V open to allow others to reproduce our work and have uploaded it as a JSON file on Zenodo, and share the link in Ref. [87].

The referee writes: “Order reflecting appearance, not importance: 1- The "for the first time" in the abstract is a claim to priority that in my opinion is neither necessary nor appropriate. I thus recommend removing it. ”

Our responses: We agree with the referee that this is no longer necessary and appropriate. We have removed it.

The referee writes: “2- Attributions in section II are a bit strange. Firstly, the original articles [34,35] are only cited at the very end of this section. Secondly, a matrix-product representation of the AKLT state appears already in the review article H.-J. Mikeska and A.K. Kolezhuk, One-Dimensional Magnetism, Lect. Notes Phys. 645, 1-83 (2004) that predates Ref. [65]. ”

Our responses: We thank the referee for pointing out the issues regarding the citations in Section. II. We have looked at the review article and it is relevant to us, as they proposed the matrix product representation of the AKLT state before Ref. [65]. We have moved the original AKLT articles (Ref. [34, 35]) to the initial part of the section, and we have included the citation of the review paper.

The referee writes: “3- The first line of Eq. (6) could be read either as | + ⟩ ( ⟨ + | + | O ⟩ ) ( ⟨ O | + | − ⟩ ) ⟨ − | or as ( | + ⟩ ⟨ + | ) + ( | O ⟩ ⟨ O | ) + ( | − ⟩ ⟨ − | ) . I recommend adding brackets in order to avoid any possible confusion. ”

Our responses: We thank the referee for mentioning this to us. Indeed Eq. (6) can be misleading without the brackets. We have added brackets to avoid the confusion.

The referee writes: “4- Beginning of section III C: the authors say that the default transpile function from Qiskit yields 24 C X gates to represent the unitary ^ U . Just before, they say that a circuit with more than a few hundred C X gates would not be ideal. I think that this combination of the two statements is confusing, and recommend clarifying this discussion. ”

Our responses: We are grateful to the referee for pointing this out. This has also been mentioned by another referee in his report as well. We have changed ‘…a few hundred CX gates…’ to ‘…a few tens CX gates…’ in our revised manuscript.

The referee writes: “5- In relation to Fig. 3: the authors should make their optimized circuit ^ V accessible. ”

Our responses: We thank the referee for his/her suggestion. We have made the dataset for the optimized circuit open and have uploaded on Zenodo and share the link in Ref. [87].

The referee writes: “6- Fig. 4: the color scales were too small to be readable in my printout. ”

Our responses: We have enlarged the scales in the color bar in Fig. 4.

The referee writes: “7- As far as I can see, Refs. [33,49,53-55,110,117,133] have been published in the meantime. I do realize that at least in some cases, the references were still unpublished when the present work was submitted, but I still recommend that the authors use the opportunity to update their references when they resubmit. ”

Our responses: Thanks for pointing out this, and we have updated the mentioned references to their published formats.

The referee writes: “8- Remove the spurious "july 2011 Special Issue" from Ref. [46].

9- Ref. [70] is a duplicate of Ref. [65].

10- Ref. [103] lacks a DOI. ”

Our replies to 8, 9, and 10: After we change the format to SciPost Physics, we have thoroughly polished the format of the references so that they are all consistent with each other. We have removed the duplicated Ref. [70]. We have added the DOI for Ref. [103] as well as all the other peer-reviewed references which lack the link of DOI.

The referee writes: “11- Remove the "in Sec. C2" above Fig. 10 (this is an unnecessary self-reference of appendix C2 to itself). ”

Our responses: We have removed this self-reference of appendix C2.

The referee writes: “12- Fig. 11: the color scales were too small to be readable in my printout. ”

Our responses: We have enlarged the scales in the color bar in Fig. 11.

The referee writes: “13- It would be nice if the authors could use the appropriate SciPost Physics style files when they resubmit”

Our responses: Thanks for the timely suggestion. We have already done so.

---

## Editorial Decision

resubmitted